# Assessment of the Manoeuvrability Characteristics of a Twin Shaft Naval Vessel Using an Open-Source CFD Code

Andrea Franceschi [1],*, Benedetto Piaggio [1], Roberto Tonelli [2], Diego Villa [1] and Michele Viviani [1]

1   Department of Electrical, Electronic and Telecommunications Engineering and Naval Architecture (DITEN), University of Genova, 16126 Genova, Italy; benedetto.piaggio@unige.it (B.P.); diego.villa@unige.it (D.V.); michele.viviani@unige.it (M.V.)
2   Maritime Research Institute Netherlands (MARIN), 6708 PM Wageningen, The Netherlands; R.Tonelli@marin.nl
*   Correspondence: andrea.franceschi@edu.unige.it; Tel.: +39-348-884-8367

**Abstract:** The purpose of this study is to assess the quality of the manoeuvre prediction of a twin-shaft naval vessel by means of a time-domain simulator based on Computational Fluid Dynamics (CFD) hydrodynamic coefficients. The simulator uses a modular approach in which the hull, rudders, appendices and propellers are based on different mathematical models. The hydrodynamic coefficients of the hull in the bare and appended configurations are computed using virtual captive tests performed with an open-source CFD code: OpenFoam. This paper demonstrates that the application of the CFD hydrodynamic coefficients led to a good estimate of the macroscopic characteristics of the main IMO manoeuvres with respect to the experimental measures. The adopted test case is the DTMB 5415M frigate both with and without appendages. This test case has been investigated in several research studies and international benchmark workshops, such as SIMMAN 2008, SIMMAN 2014 and many CFD workgroups.

**Keywords:** reynolds average navier-stokes (RANS); DTMB 5415; manoeuvrability; hydrodynamic coefficients; virtual captive tests

## 1. Introduction

Early prediction of the manoeuvring abilities has become a primary aspect of the ship design process. Since 2003, when International Maritime Organization (IMO) requirements were promulgated [1,2], the necessity of a better and earlier prediction of the manoeuvrability of the ship has become a driving aspect of the ship design. Regulations provide guidelines and limits regarding merchant ship characteristics, while naval vessels have even more challenging design requirements. In both cases, a precise and reliable prediction of the manoeuvring characteristics of the ship is required.

Model tests are the most reliable methodology to predict the manoeuvrability of the ship. A significant number of studies have been conducted concerning the prediction of manoeuvrability through both captive model tests and free model tests [3–6]. Strip theory, semi-empirical models and statistical regressions have been formulated and validated over the last decades and they have proven to be a particularly simple and time-saving approach for early assessment of a ship's manoeuvring characteristics [7–9]. However, as pointed out by Viviani et al. [10], these regressions are often derived from the analysis of single-screw ships.

The use of these methods does not provide results with a satisfactory level of accuracy for twin shaft naval vessels, when the appendage configuration is decisive. Computational Fluid Dynamics (CFD) calculations represent an important tool to predict the manoeuvring ability of the ship in different applications [11–13]. The usage of CFD methods in this field can be summarized in two groups, namely: free running numerical approach, which leads to satisfactory results with high time consuming simulations [14,15] and virtual captive

model tests, which is the most widespread adopted method [16,17]. In the latter case, the numerical hydrodynamic coefficients are used to feed a time-domain simulator, which predicts the whole ship dynamic.

In the present study, we assess the quality of the second methodology considering two aspects. First, a validation of the CFD captive tests results by comparing numerical results with experimental data. Then, we predict the ship manoeuvring characteristics by comparing simulator results with measurements obtained by means of free running model tests.

This study, despite being conducted on a well known subject, offers an interesting reflection and presents some original features in the calculation of the ship's hydrodynamic derivatives. First, the CFD calculations are performed with a steady solver, which drastically reduces the calculation time of each CFD simulation with respect to similar studies conducted with an unsteady solver [18]. This shortening of the required computational time makes this approach more attractive from a day-by-day application standpoint.

Unlike studies in which the calculations are stationary and the attitude of the ship is kept constant during the test [19], in this work, the CFD calculations take into account the current attitude of the ship by considering the actual numerical result. This modification of the ship attitude is done through an algorithm based on the metacentric hypothesis. The test case of this paper is the US Navy combatant DTMB 5415, also known as 5415 M. Since captive model tests are available with and without appendages, both the setups are investigated numerically and compared with experiments.

CFD calculations were carried out to compute the hydrodynamic coefficients related to the hull (bare hull calculations) and hull + appendages contributions (fully appended calculations). The rudder and propeller forces (and the appendages force when bare hull hydrodynamic coefficients are adopted) are added by means of semi-empirical models developed at Universitá Degli Studi di Genova (UNIGE) in recent years [10].

This paper starts with an overview of the main models on which we based the manoeuvre simulations. This is followed by a detailed description of the numerical setup used in the CFD calculations. Section 4 reports the comparison between the numerical and experimental results in terms of the global and local forces acting on the ship in the captive tests. In Section 5, the hydrodynamic coefficients computed from the previous sections are used to simulate the manoeuvres. These simulations are compared with the time traces of the free running tests. In the last section, the results obtained through this study are condensed, emphasizing both the potential of the methodology and its critical issues.

## 2. Manoeuvrability Model

The forces and the velocities presented in the following chapters are referred to the ship fixed reference frame. The origin is located at midship with the x-direction pointing from stern to ship bow and the z-direction downwards as reported in Figure 1.

Forces and moments are made non-dimensional according to the formulas reported in the system of Equation (1).

$$
\begin{cases}
F' = \dfrac{F}{(0.5\rho V^2 L_{PP}^2)} \\
M' = \dfrac{M}{(0.5\rho V^2 L_{PP}^3)} \\
F'_{rudder} = \dfrac{F_{rudder}}{(0.5\rho V^2 s_R c_R)}
\end{cases}
\tag{1}
$$

Regarding the time-domain simulator, this paper adopts a modular approach [20,21], which consists of an independent description of the different force and moment components acting on the ship during the manoeuvre. The hull, rudders, appendices and propellers are modelled separately, and correlation factors are taken into account in each module.

The simulator adopts a three DOF model in which the surge, sway and yaw velocities are correlated by the system of Equation (2), while the roll motion is not considered here.

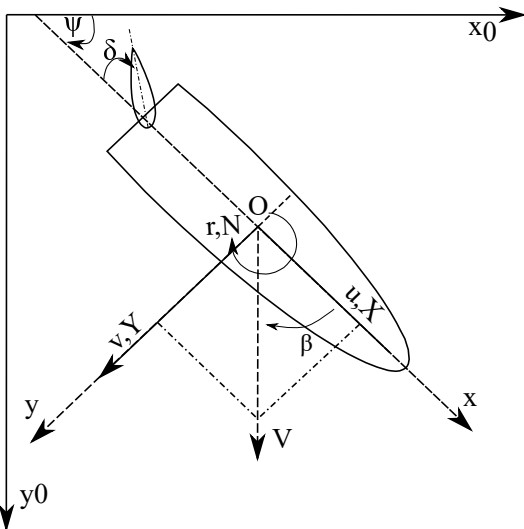

**Figure 1.** Manoeuvrability sign convention.

Figure 2 shows the estimated roll angle during a turning circle $\delta = 35°$ manoeuvre performed using the strip theory mathematical model in the time domain simulator. This figure shows that the roll angle in the turning circle manoeuvre (performed at $V_s$ equal to 18 knots) reached about 3 degrees. Therefore, the influence of the roll motion can be considered negligible on the in-plane forces and moments.

$$\begin{cases} X = (\Delta - X_{\dot{u}})\dot{u} - \Delta(v + rx_G)r \\ Y = (\Delta - Y_{\dot{v}})\dot{v} + \Delta x_G\dot{r} + \Delta ur \\ N = (I_{zz} - N_{\dot{r}})\dot{r} + \Delta x_G\dot{v} + \Delta ux_G r \end{cases} \tag{2}$$

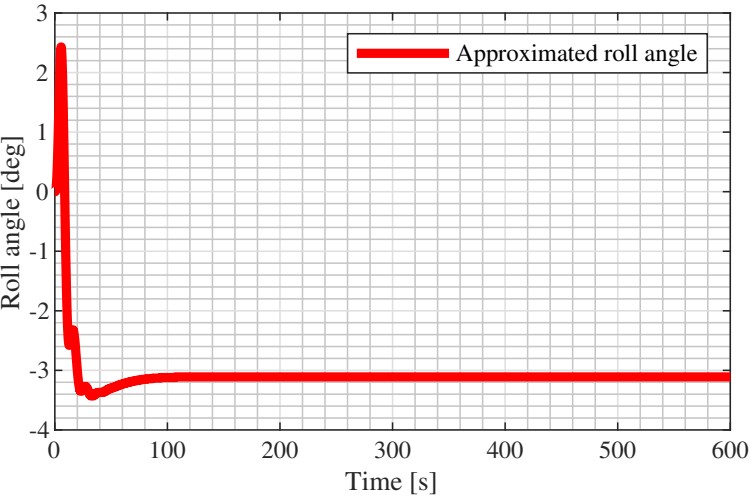

**Figure 2.** Roll angle in turning circle 35 [deg] manoeuvre.

The global forces acting on the ship are decomposed considering the different contributions as reported in the system of Equation (3).

$$\begin{cases} X = X_{hull} + X_{rudders} + X_{appendices} + X_{propellers} \\ Y = Y_{hull} + Y_{rudders} + Y_{appendices} + Y_{propellers} \\ N = N_{hull} + N_{rudders} + N_{appendices} + N_{propellers} \end{cases} \tag{3}$$

Hull forces are modelled by means of the system of Equation (4).

$$\begin{cases} X_H = R(u) + X_{vv}v^2 + X_{rr}r^2 + X_{vr}vr \\ Y_H = Y_0 + Y_v v + Y_{v|v|}v|v| + Y_{vvv}v^3 + Y_r r \\ \quad + Y_{r|r|}r|r| + Y_{vvr}v^2 r + Y_{vrr}vr^2 \\ N_H = N_0 + N_v v + N_{v|v|}v|v| + N_{vvv}v^3 + N_r r \\ \quad + N_{r|r|}r|r| + N_{vvr}v^2 r + N_{vrr}vr^2 \end{cases} \tag{4}$$

This work focuses on the hull and appendages forces; thus, all the other hydrodynamic devices are described by mathematical models calibrated on literature results and/or experiments. The details of the rudder, propeller and appendages mathematical models implemented in the time-domain simulator are not discussed in this study.

For the sake of completeness, a concise outline of the characteristics of the rudder, propeller and appendages models is given below. Regarding the rudder forces, the simulator computes the lift and the drag by means of a mathematical model based on theory proposed by Molland et al. [22]. The model includes the rudder geometry, type and the interactions with the hull and the propeller. The rudder forces mathematical model is designed to work in the whole [0,2π] range of angles of attack and it is validated against the experimental data available in the literature [23]. The propeller-rudder interaction is described by a model based on Abkowitz theory [24]. The range of angle of attack in which the rudder operates during standard manoeuvres is divided in three zones: linear (1), pre-stall (2) and post-stall (3) as reported in Figure 3.

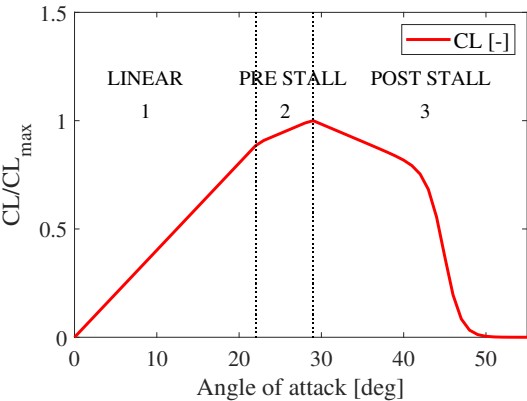

**Figure 3.** Lift coefficient of the rudder implemented in the simulator.

The propeller model computes both the longitudinal and lateral forces due to the oblique flow at the propellers. The thrust and torque coefficients of the propeller are computed by interpolation on the geometric characteristics of the actual propeller by the Wageningen B-series experimental results [25], while the model of in-plane forces is based on the approach proposed by Ribner [26].

The appendages contribution is based on the theory proposed by Jacobs [27] and calibrated over the experimental results.

## 3. Numerical Setup

This chapter illustrates the main characteristics and the numerical setup of the simulations performed on the 5415 model in the bare hull configuration. The structure of the mesh is level based: the computational domain is created and modelled with the *blockMesh* tool, which is a mesh generator provided by the OpenFoam suite.

The mesh is refined by the definition of several blocks around the ship. The largest one is the whole computational domain whose main dimensions are reported in Table 1 and, graphically, in Figure 4. The overall domain is modelled by the *blockMesh* OpenFoam command, while the smallest block is modelled separately by the usage of the *CfMesh*

library. This is due to the superior ability to handle complex geometries with *CfMesh* with respect to the standard mesh generator libraries adopted in OpenFoam. The dimensions of the overall domain are automatically computed as a function of the ship's length. The computational domain is discretized with an hex-dominant Cartesian mesh made of several mesh refinements as shown in Figures 5 and 6.

**Table 1.** OpenFoam domain dimensions.

| Computational Domain Dimensions | |
|---|---|
| $L_X$ [$L_{PP}$] | 4.5 |
| $L_Y$ [$L_{PP}$] | 2.5 |
| $L_Z$ [$L_{PP}$] | 2 |

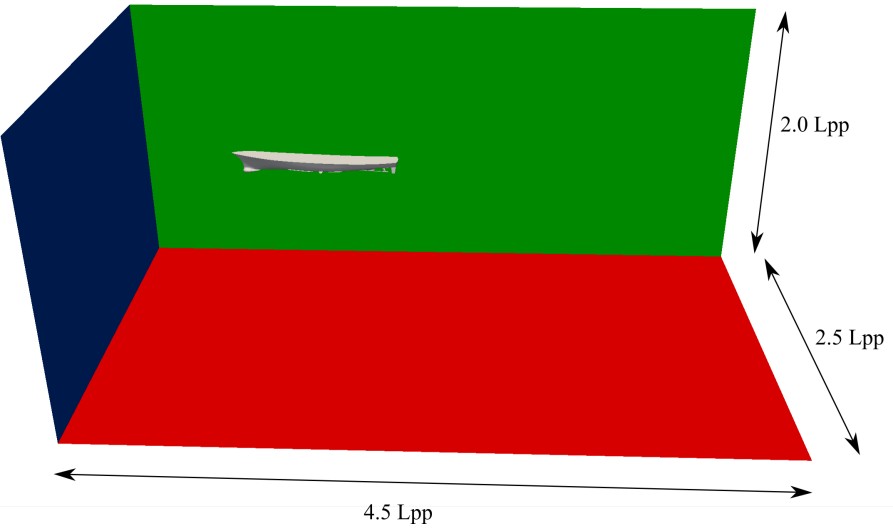

**Figure 4.** Computational domain.

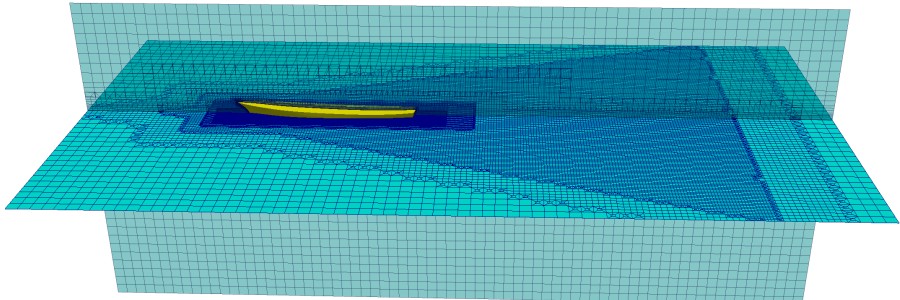

**Figure 5.** Mesh in the pure drift test.

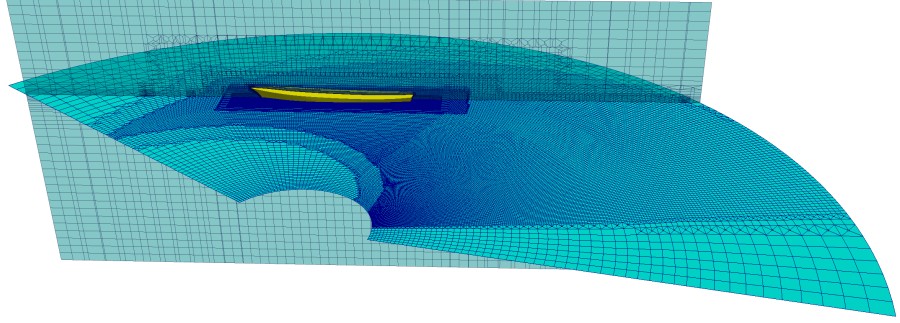

**Figure 6.** Mesh in the rotation test.

The free surface and the Kelvin sector are modelled by anisotropic refinements to capture the wave propagation behind the hull. The mesh generation process distinguishes between simulations with and without the rotation rate: in the former case, the computational domain is deformed to guarantee the correct rotation of the flow around the ship. In addition to the mesh differences, the rotation tests require different numerical setups in terms of the field function definition (relative flow velocity field) and solver (inertial solver as described in the following of this section).

Figures 5 and 6 give an overview on the differences between the mesh setup in the drift and rotation tests. In particular, Figure 6 represents the deformation of the computational domain when a rotation rate is considered. These figures show that, in pure drift tests, the computational domain is a parallelepiped, while in rotation tests, it is defined as a truncated cone. The curvature of the cone is computed such that the rate of turn at midship is equal to the rotation rate of the test.

The CFD calculations are performed using a quasi steady time approach (named the Local Time Stepping approach) implemented in the *InterFoam* solver. The steadiness of the adopted approach does not permit to consider the dynamic change of the trim and the sinkage of the ship during the simulation. This method reduces the computational efforts of the numerical simulations. Nevertheless it introduces the critical issue of changing the current attitude of the simulation to consider the dynamic positioning of the ship during the test.

Consequently, the dynamic trim and sinkage of the ship are computed by iteratively modifying the running attitude in order to guarantee the balance between the total vertical force and the pitch moment acting on the hull. The simulation is divided in a specified number of steps, further characterized by a fixed number of iterations. The number of steps is chosen in order to guarantee the balance between vertical forces and pitch moments at the end of the simulation while the number of iterations is chosen in order to guarantee an acceptable convergence of the main flow quantities.

At the end of each step, the vertical force and pitch moment acting on the ship are evaluated, and the running attitude is modified. Figures 7 and 8 show the convergence of the vertical force and the pitch moment (red lines) due to the modified attitude versus the external forces (black lines). After the first 6000 iterations, the average value of Z and M are evaluated, and the trim and sinkage are updated considering equations based on the metacentric hypothesis. This algorithm takes as input data the waterline area, the volume, the longitudinal metacentric radius and the forces formerly defined and it computes the trim and the sinkage of the ship.

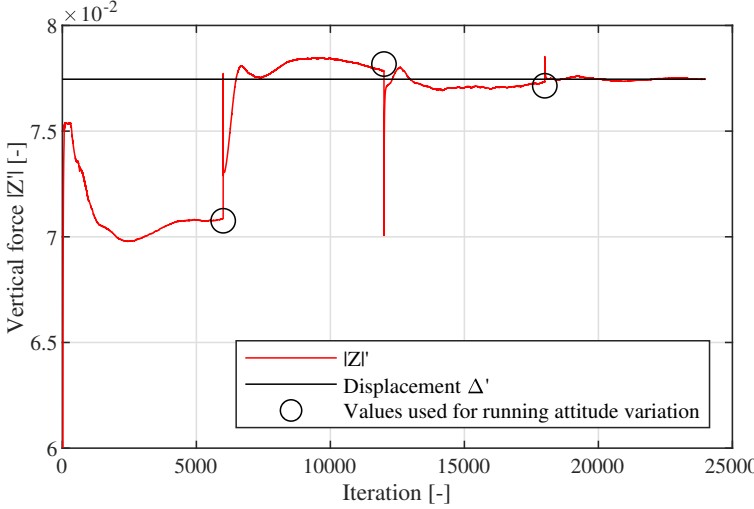

**Figure 7.** Iterative process of the vertical force balance.

This process is iterated at least three times, allowing the system to reach an equilibrium of forces and moments, as shown in Figures 7 and 8. The number of iterations is chosen

such that a satisfactory balance between the forces and moments is obtained at the end of the simulations. The difference between the vertical force and the pitch moment computed by OpenaFoam and that expected from the loading condition of the model was less than 5%, which is considered a satisfactory result. This process guarantees evaluation of the hydrodynamic forces acting on the hull, also considering the ship attitude, with a reduced computation time (about 72 h on a 24-core infrastructure for full appended calculations).

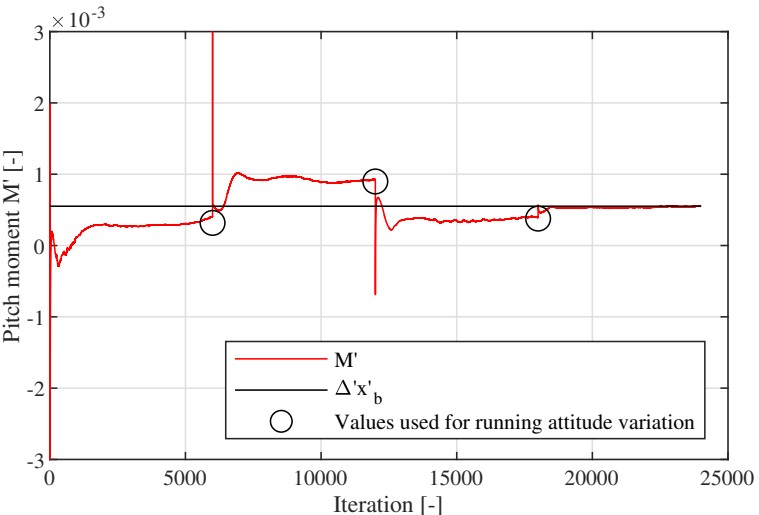

**Figure 8.** Iterative process of the pitch moment balance.

The running attitude was found to be crucial in order to correctly predict the hydrodynamic coefficients. Variations of more than 20% were obtained on the horizontal forces and moments during shifting between the inital condition and the trimmed state. The Richardson mesh convergence approach, as proposed by Celik [28], was applied on the CFD calculations to deduce the appropriate mesh density of the numerical campaign.

Mesh Independence Analysis provides information about the mesh density required to obtain acceptable numerical results; the analysis was conducted on both the drift and rotation simulations. Table 2 shows that the dynamic quantities (X, Y, N), which are used to model the manoeuvrability of the ship, converge with the increase of the mesh density. The maximum error between the extrapolated and standard mesh values is related to the yawing moment in the pure drift test and it is about 3%, which is considered acceptable.

The computation times for the simulations, performed with a 24-cores infrastructure of the bare hull setup on the coarse, standard and fine grids were about 18, 43 and 95 h, respectively. Based on these analyses, the best compromise between computational effort and accuracy was achieved by assuming a network with 1.5 M cells, whose main data are listed in Table 3.

As already mentioned, the solver adopted in numerical tests without the rotation rate is *LTSInterFoam*, which is a multi-phase quasi-steady solver implemented in OpenFoam that relies on the Local Time Step (LTS) method. The LTS method manipulates each individual cell time step to reach as soon as possible the steady state condition of the simulation. The rotational and combined tests use a different solver that also takes into account the inertial components due to a rotation rate. This solver is the *SRFInteFoam*, which allows consideration of the correct velocity in a non-inertial rotating reference frame (Single Rotating Frame).

The whole simulation campaign was conducted considering the widely adopted $k$-$\omega$ *SST* turbulence model initially proposed by Menter [29]. The main numerical schemes used in the simulations are reported in Table 4.

**Table 2.** Richardson analysis data.

| Richardson Mesh Convergence Analysis | | | |
|---|---|---|---|
| Mesh | Fine | Standard | Coarse |
| $\lambda$ | $1/\sqrt{2}$ | 1 | $\sqrt{2}$ |
| h | 1.77 | 1.59 | 1.42 |
| $r_{21}$ | | 0.90 | |
| $r_{32}$ | | 0.89 | |
| Drift test | | | |
| Longitudinal force X' | | | |
| Value | $-1.35 \times 10^{-3}$ | $-1.36 \times 10^{-3}$ | $-1.80 \times 10^{-3}$ |
| Convergence | | Monotonic | |
| $GCI_{fine}^{21}$ | | 0.3 | |
| Transverse force Y' | | | |
| Value | $7.11 \times 10^{-3}$ | $6.98 \times 10^{-3}$ | $6.95 \times 10^{-3}$ |
| Convergence | | Monotonic | |
| $GCI_{fine}^{21}$ | | 0.62 | |
| Yawing moment N' | | | |
| Value | $2.81 \times 10^{-3}$ | $2.86 \times 10^{-3}$ | $2.89 \times 10^{-3}$ |
| Convergence | | Monotonic | |
| $GCI_{fine}^{21}$ | | 2.2 | |
| Rotation test | | | |
| Longitudinal force X' | | | |
| Value | $-1.07 \times 10^{-3}$ | $-1.08 \times 10^{-3}$ | $-1.09 \times 10^{-3}$ |
| Convergence | | Monotonic | |
| $GCI_{fine}^{21}$ | | 5.0 | |
| Transverse force Y' | | | |
| Value | $-2.05 \times 10^{-3}$ | $-2.03 \times 10^{-3}$ | $-1.92 \times 10^{-3}$ |
| Convergence | | Monotonic | |
| $GCI_{fine}^{21}$ | | 0.4 | |
| Yawing moment N' | | | |
| Value | $-1.43 \times 10^{-3}$ | $-1.44 \times 10^{-3}$ | $-1.49 \times 10^{-3}$ |
| Convergence | | Monotonic | |
| $GCI_{fine}^{21}$ | | 0.1 | |

**Table 3.** Mesh and fluid characteristics in the bare hull calculations.

| Mesh Characteristics | |
|---|---|
| Description | cartesian |
| Type of grid | unstructured |
| Number of cells | 1.5 M |
| y+ on the hull | 15 |
| Stretching ratio | 1.2 |
| Number of surface elements | 200 k |
| **Fluid properties** | |
| Density [kg/m$^3$] | 1000 |
| Kinematic viscosity [m$^2$/s] | $1 \times 10^{-6}$ |
| Reynolds number | $4.3 \times 10^5$ |
| Froude number | 0.280 |

**Table 4.** Numerical schemes.

| Numerical Schemes | |
|---|---|
| ddtSchemes | localEuler |
| gradSchemes | Gauss linear |
| divSchemes | Gauss linearUpwind |
| laplacianSchemes | Gauss linear |
| interpolationSchemes | linear |
| snGradSchemes | limited |

The setup used for generating the mesh in the appended hull configuration is the same as that presented above. Figure 9 gives an overview of the surface mesh quality on the aft of the ship in the appended setup. Additional refinements are used to describe the appendage geometries. The mesh generation and the numerical setup adopted for the appended hull configuration are the same as for the bare hull configuration, however, with a larger number of cells, as reported in Table 5. This selection was obtained by performing a mesh-dependence analysis as before.

The presence of the propellers is described by an actuator disk model, which provides an acceleration effect on the flow by the adoption of body forces. The actuator disk model was developed by UNIGE, and it has been extensively validated in previous studies [30,31]. The thrust and torque radial distribution are taken as input for the actuator disk model. The radial distributions are obtained by the cited studies on the propeller–rudder interaction and they are reported graphically in Figure 10. The values of total thrust and torque developed by the propellers are forced equal to MARIN captive tests measurements.

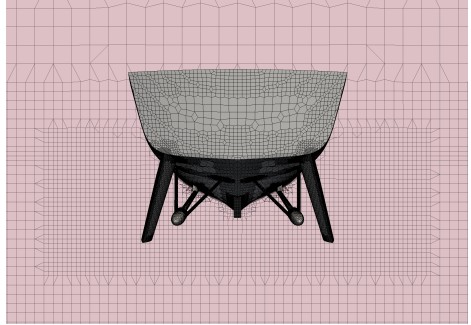 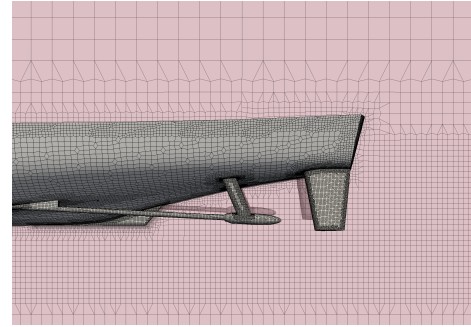

**Figure 9.** Refinement of the surface mesh on the aft of the ship.

This model guarantees to correctly include the propeller acceleration effect with a negligible increase of the computational burden.

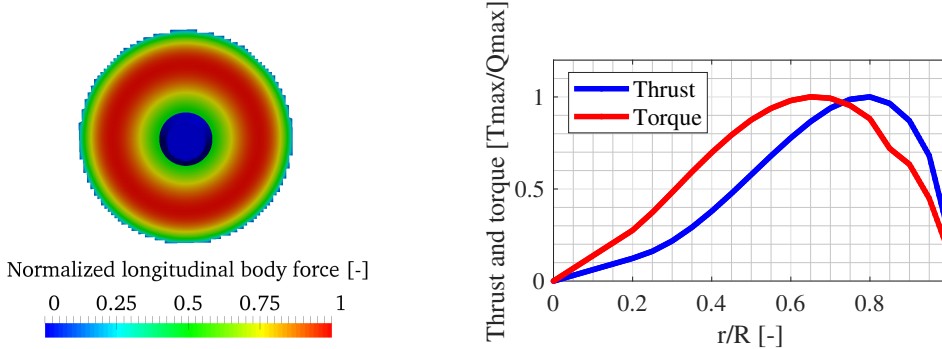

**Figure 10.** The actuator disk body force radial distribution.

**Table 5.** Mesh characteristics in the appended calculations.

| Mesh Characteristics | |
|---|---|
| Description | cartesian |
| Type of grid | unstructured |
| Number of cells | 5.3 M |
| y+ on the hull | 1.6 |
| Stretching ratio | 1.2 |
| Number of surface elements | 300 k |

## 4. CFD and EFD Captive Tests

### 4.1. Test Case

DTMB 5415 is a documented hull form that is publicly available in the literature. This test case, in its bare hull configuration, was used in the past for several hydrodynamic studies regarding ship resistance [32,33] wake fields and manoeuvring forces [34]. In 1999, a fully-appended configuration was conceived and successively made public in 2006. In the same years, self-propelled manoeuvring and sea-keeping tests [35] were performed and shared. That arrangement of the appended DTMB 5415 is indicated as 5415 M.

Figure 11 shows a picture of the fully-appended model. Table 6 contains the main characteristics of the ship in full scale. Free running model tests were conducted by MARIN at ship speed of 18 and 30 knots on the 5415 M model. The present study investigates the manoeuvring ability of the vessel at a speed equal to 18 knots. The FORCE and CNR-INM (formerly INSEAN) institutes performed captive tests at different ship velocities: the experimental data considered in this study are the ones referred to a ship speed of 20 knots.

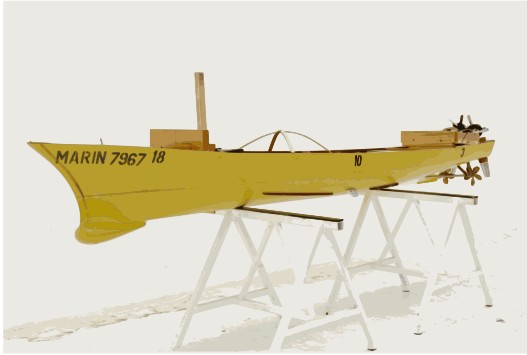

**Figure 11.** The 5415 M model in MARIN facilities.

The main geometric characteristics of the models used during the EFD test campaigns are reported in Table 7. The 5415 M model was equipped with two propeller shaft lines supported by V-shape brackets, two spade rudders, a small skeg and two bilge keels

interrupted by a fin stabilizer for each side. The main characteristics of the appendices, rudders and propellers are reported in Tables 8–10.

**Table 6.** The DTMB 5415 test case.

| Test Case | | |
|---|---|---|
| **Description** | **Unit** | **Value** |
| $L_{PP}$ | m | 142.0 |
| B | m | 19.0 |
| T | m | 6.14 |
| $x_B$ | m | 71.60 |
| $V_S$ | kn | 18.0 |
| Fr | - | 0.248 |

**Table 7.** The model characteristics.

| Characteristics of the Models | | | | |
|---|---|---|---|---|
| Quantity | SHIP | INSEAN | FORCE | MARIN |
| $\lambda$ [-] | 1 | 24.83 | 35.48 | 35.48 |
| $L_{PP}$ [m] | 142 | 5.72 | 4.00 | 4.00 |
| B [m] | 19 | 0.768 | 0.537 | 0.537 |
| T [m] | 6.14 | 0.248 | 0.173 | 0.173 |
| $\nabla$ [m$^3$] | 8.5k | 0.554 | 0.189 | 0.189 |
| Fr [-] | - | 0.138, 0.280, 0.410 | 0.138, 0.280, 0.410 | 0.248, 0.410 |
| $V_S$ [kn] | - | 10.0, 20.0, 30.0 | 10.0, 20.0, 30.0 | 18.0, 30.0 |
| Configuration | - | 5415 | 5415 | 5415 M |

**Table 8.** The geometric characteristics of the appendices.

| Geometric characteristics of the appendices | |
|---|---|
| Ship scale | |
| Rudder headbox | |
| Span [m] | 1.2 |
| Chord [m] | 4.4 |
| Skeg | |
| Lateral area [m$^2$] | 12.3 |
| Shaft | |
| Length [m] | 24.1 |
| Diameter [m] | 0.55 |
| Internal brackets | |
| Span [m] | 4.0 |
| Chord [m] | 1.3 |
| External brackets | |
| Span [m] | 3.3 |
| Chord [m] | 1.4 |

**Table 9.** The geometric characteristics of the rudders.

| Rudders | |
|---|---|
| **Ship scale** | |
| $s_R$ [m] | 4.4 |
| $c_R$ [m] | 3.5 |
| $t_{Rmax}$ [m] | 0.89 |
| Point of max. thickness [%] | 25 |
| $A_R$ (each) [m$^2$] | 15.4 |
| Total rudders area ratio [%] | 3.62 |
| Angle in Y-Z plane [deg] | 15 |

**Table 10.** The geometric characteristics of the propellers.

| Propellers | |
|---|---|
| **Ship scale** | |
| $D_P$ [m] | 6.15 |
| $P_{0.7R}$ [m] | 5.35 |
| $P/D_{0.7R}$ [-] | 0.87 |
| Boss diameter ratio [-] | 0.347 |
| $A_E/A_0$ [-] | 0.580 |
| $Z_P$ | 5 |
| Direction of rotation | Inward over the top |

The rotational and combined experimental tests reported in Table 11 were performed dynamically. Therefore, the related forces were computed by means of the mathematical models provided in the institutes technical reports.

Although institutes performed captive tests at different Froude numbers (as shown in Table 7), in the following tables, we report the test matrices related to a ship speed equal to 18/20 knots. Table 12 gives an overview of the main characteristics of the MARIN captive tests campaign.

**Table 11.** Experimental test matrix on 5415.

| EFD Test Matrix on 5415 Model | |
|---|---|
| Fr = 0.280 [-] | |
| Drift angle [deg] | Non dimensional yaw rate r' [-] |
| Static drift tests | |
| 0, 2, 6, 9, 10, 11, 12, 16, 20 | 0 |
| Rotation tests | |
| 0 | 0.05, 0.15, 0.2, 0.3, 0.45, 0.6 |
| Drift and rotation tests | |
| 9, 10, 11 | 0.3 |

**Table 12.** Experimental test matrix on 5415 M.

| EFD Test Matrix on 5415 M Model | |
| --- | --- |
| Fr = 0.280 [-] | |
| $\beta$ [deg] | $r'$ [-] |
| Static drift tests | |
| 0, 2, 4, 6, 10, 12, 16, 20 | 0 |
| Rotation tests | |
| 0 | 0.1, 0.2, 0.3, 0.45, 0.6 |
| Drift and rotation tests | |
| 6/10 | 0.2, 0.3/0.1, 0.2, 0.3 |

*4.2. Bare Hull Calculations*

This subsection reports the comparison between the CFD results and experimental measurements on the 5415 model (bare hull configuration). As the first step, the capability of the proposed method to capture the running attitude during the tests was checked. The experiments provided data about the trim and the sinkage of the ship measured in pure drift static tests. Positive sinkage means that the draft increased, and positive trim means that the ship trims by the bow.

In Figures 12 and 13, the numerical results in terms of the trim and sinkage are compared with experimental values. These figures present the measurements related to the two institutes separately, in order to show the experimental uncertainty on the kinematic quantities. INSEAN measurements are named EFD1, while FORCE measurements are named EFD2.

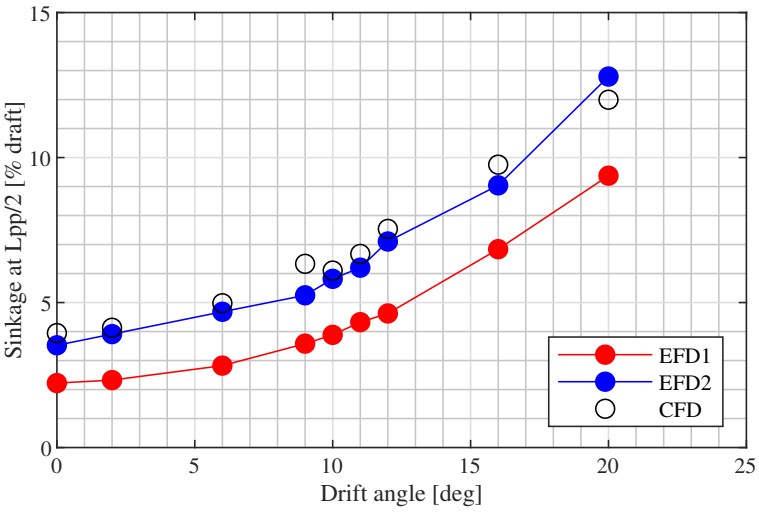

**Figure 12.** Experimental vs. numerical sinkage at midship.

The CFD results show a good agreement with the experimental measurements, and both the sinkage and trim trends are correctly predicted; as it can be seen, only a small gap exists for the sinkage at midship for one institute measurement, while, for the other one, the sinkage is perfectly predicted. In the former case, the CFD calculations over-estimate the sinkage, with differences around 1.5–2.5 mm (1–2% of the draft).

This over-prediction of the dynamic sinkage can be due to the transducer precision limit, which is around 1 mm (as regulated by ITTC guidelines [36]). Finally, Figure 13 shows a good agreement between numerical trim and the measurement reported by the first institute: the trend is well described, while the over-estimation of the trim value is

around 0.15 deg. The second institute measurement shows a considerable over-estimation of the trim obtained by the CFD calculations.

On the other hand, this lack of description is related to a very small quantity (with respect to the total draft), and thus it is supposed to have a slight influence on the global forces acting on the ship during the tests. The two institutes provide trim and sinkage measurements with substantial differences, and the CFD calculations are considered to be acceptable by considering the experimental uncertainty. The following figures report the comparison between the experimental and numerical results in terms of the lateral force, yawing moment and destabilizing lever in the pure drift tests. The experimental results are reported as the averaged value of the measurements provided by the two institutes.

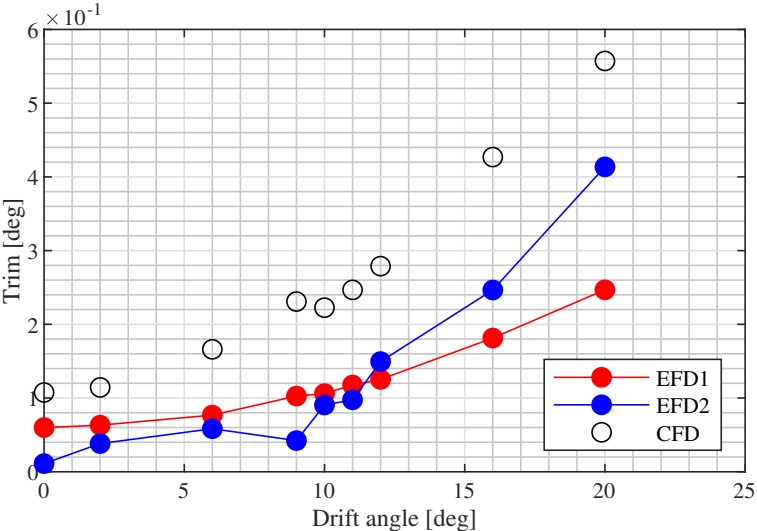

**Figure 13.** Experimental vs. numerical trim.

In Figures 14, 15, 16, 17, 18 and 19 the results in terms of the forces and moments are reported and compared with the experimental data.

Figure 14 shows a satisfactory correspondence between the numerical calculations and experimental data in terms of the lateral force. The force was slightly under estimated for the higher values of the drift angle (difference around 8%). Figure 15 shows that the numerical calculations led to a slight underestimation of the yawing moment acting on the ship, and the overall difference between numerical and experimental results was around 6%. Figure 16 describes the lack of description of the longitudinal position of the centre of pressure on the ship in pure drift tests. CFD calculations led to an over-estimation of the longitudinal coordinate of the centre of pressure with respect to the experiments.

This over-estimation was larger for small values of the drift angle, whereas the comparison shows a good numerical-experimental agreement for larger drift angles. This could be partially due to the small recorded values at small drift angles, generating an higher uncertainty in the EFD measurements. In the following section, the comparison between the numerical and experimental results related to rotation tests is discussed. For the sake of completeness, these figures report the comparison of the lateral force in rotation even if this quantity has a limited effect on the manoeuvring characteristics of the ship. Figure 17 shows the good prediction of the lateral force computed by CFD calculations with respect to the experimental results.

In Figure 18, the yawing moment acting against the yaw rate is presented, showing a very good correspondence between the numerical and experimental results. The difference between the CFD and experiments is about 6%. Figure 19 illustrates the quality of the prediction of the yawing moment in the combined drift and rotation captive tests. As can be seen, in these calculations, the small discrepancy highlighted for the pure drift tests is almost nullified. The X force in drift and yaw tests was omitted because it had a small influence on the manoeuvre behaviour.

As it can be seen, a satisfactory agreement was found, as a whole, in the drift tests with the differences slightly higher than 5%. As already discussed, a very good agreement was found for the most influential yawing moment in the rotation tests. As a whole, the CFD calculations led to a slightly higher destabilizing moment at low drift angles, which was compensated for almost completely when moving to the yaw and drift tests.

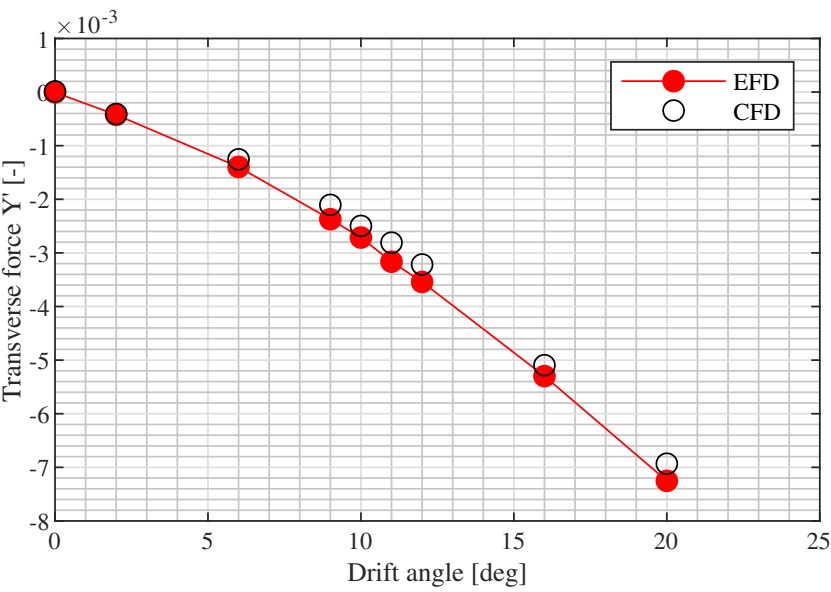

**Figure 14.** Transverse force Y′ in the static drift tests-bare hull configuration.

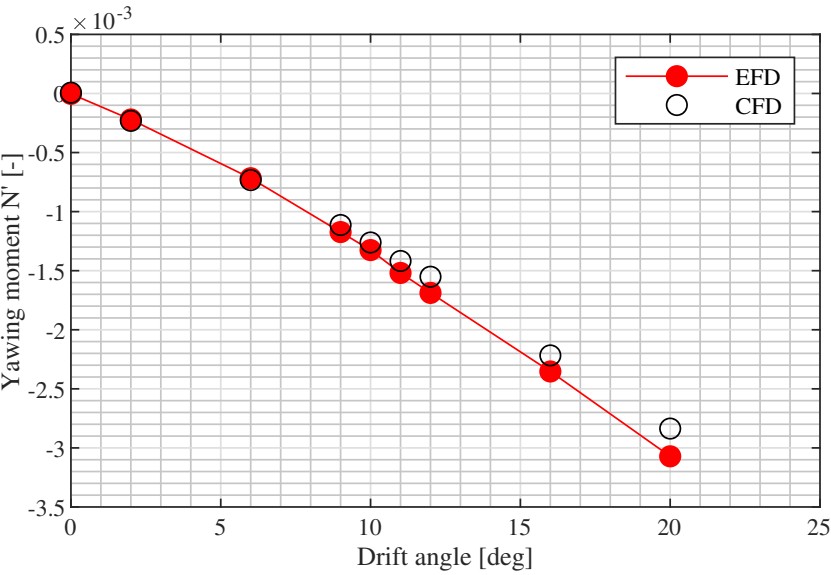

**Figure 15.** Yawing moment N′ in the static drift tests-bare hull configuration.

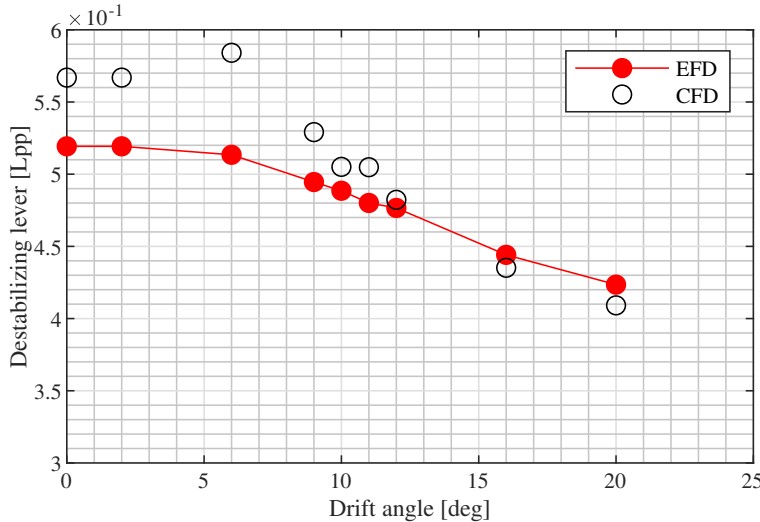

**Figure 16.** Destabilizing lever in the static drift tests-bare hull configuration.

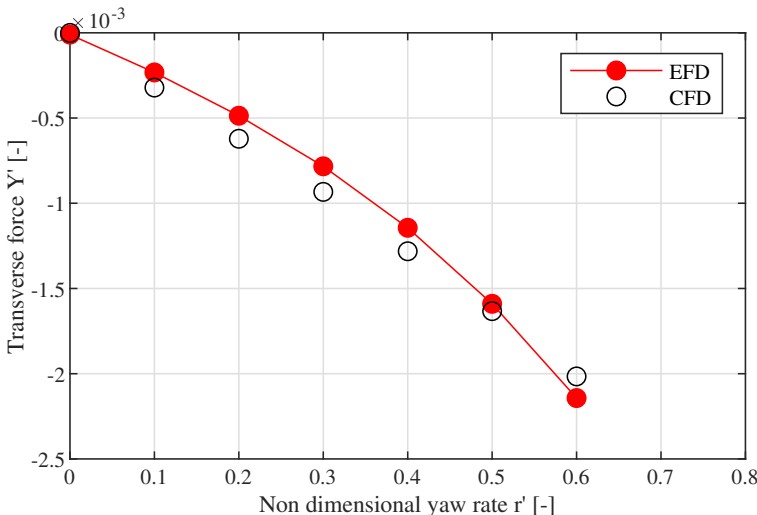

**Figure 17.** Transverse force in the static yaw rate tests-bare hull configuration.

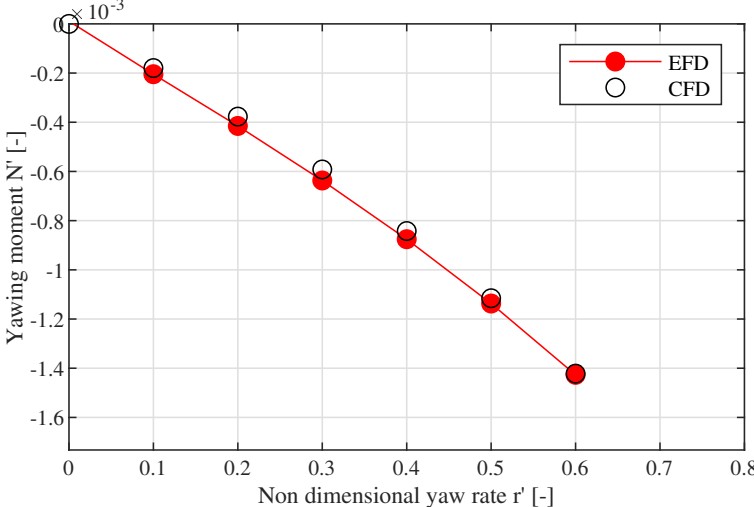

**Figure 18.** Yawing moment in the static yaw rate tests-bare hull configuration.

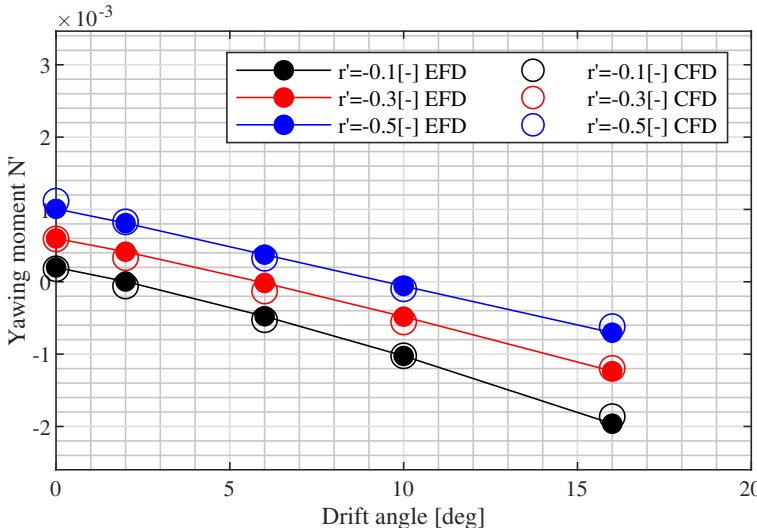

**Figure 19.** Yawing moment in the combined drift and yaw rate tests-bare hull configuration.

### 4.3. Fully Appended Calculations

As in the bare hull section, the following figures report the comparison between numerical and experimental results in terms of the transverse force, yawing moment and destabilizing lever in the pure drift tests.

Figure 20 shows that a very good agreement was present in the range of drift angles up to 10 degrees in the lateral force prediction, while this was slightly underestimated at higher values. For the yawing moment, Figure 21 shows that a better correspondence was found at higher drift angles, with a slight underestimation for intermediate values. Figure 22 shows that, as for the lateral force, CFD calculations led to a slight underestimation of the global destabilizing moment related to the drift tests.

The overall difference between the numerical results and experiments in drift tests was about 6% for both the Y force and N moment. Figure 22 shows a good estimation of the destabilizing lever for high drift angles and a general under-estimation of the lever for low drift angles. The following figures report the comparison between the CFD results and experiments in terms of the transverse force and yawing moment in rotation tests.

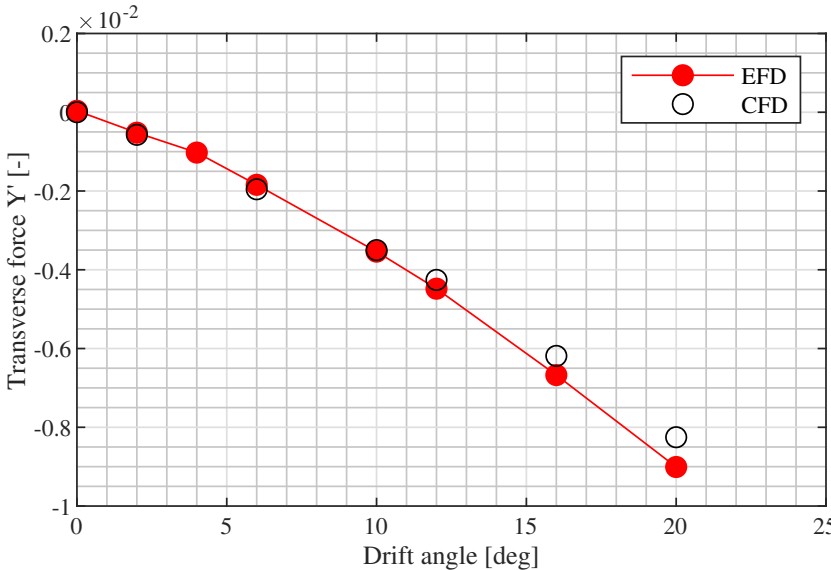

**Figure 20.** Transverse force Y′ in the static drift tests-fully appended configuration.

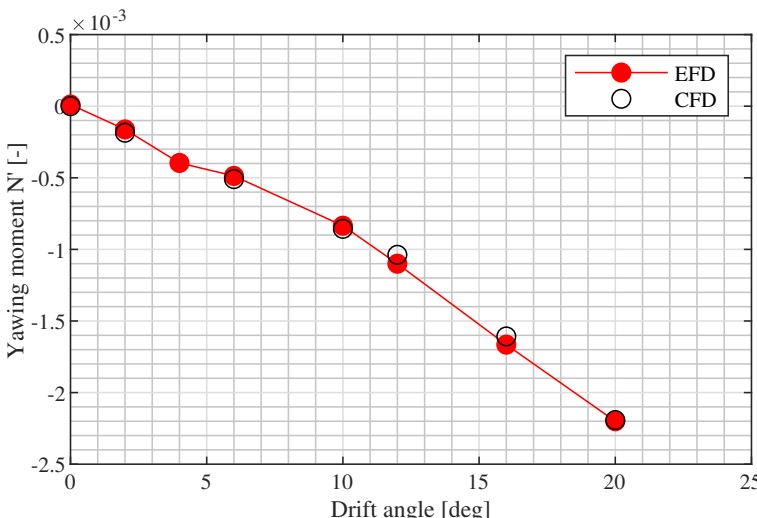

**Figure 21.** Yawing moment N' in the static drift tests-fully appended configuration.

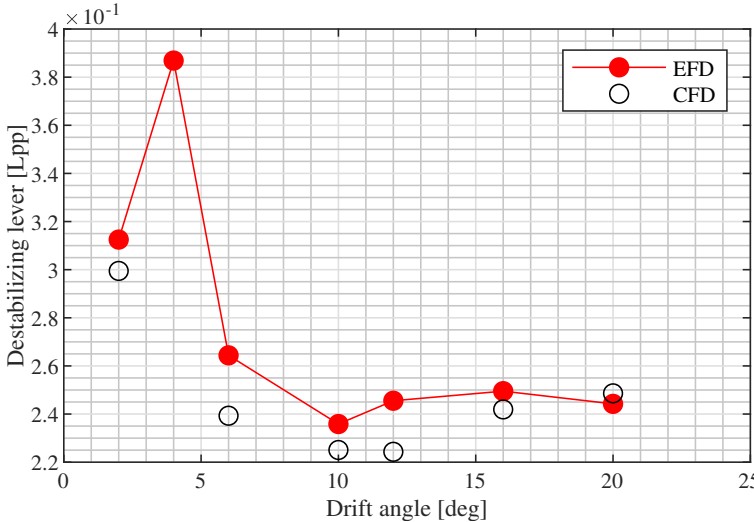

**Figure 22.** Destabilizing lever in the static drift tests-fully appended configuration.

Figure 23 shows a good agreement between the Y force up to r' = 0.3 [-] (linear range), while the lateral force was under-estimated for higher rates of turn. Nevertheless, the transverse force occurring in rotation had a weak influence on the manoeuvres. This comparison prefigures a misleading of the CFD calculations in rotation tests, which is discussed further below. In Figure 24, the yawing moment acting against the yaw rate is reported. As it can be seen, a rather good agreement was present up to 0.3 [-], while, for higher values, the moment was underestimated, leading to a more unstable ship.

This lack of description refers to an under-estimation of the lateral forces acting on the aft part of the ship, as discussed in detail in the next section. Figure 25 reports a comparison between the numerical and experimental results in the combined drift and yaw tests. This figure shows that the CFD results predicted with high accuracy the yawing moment due to a drift angle while the accuracy was considerably worsened in the case of the mixed drift and yaw tests—consistent with what was shown for the pure yaw tests.

Considering the previously reported results, these discrepancies at high angle of attacks can be ascribed to the appendage description. This will be discussed in the following section. The rotation tests results show an under-estimation of the yawing moment due to a rotation by slightly more than 15%.

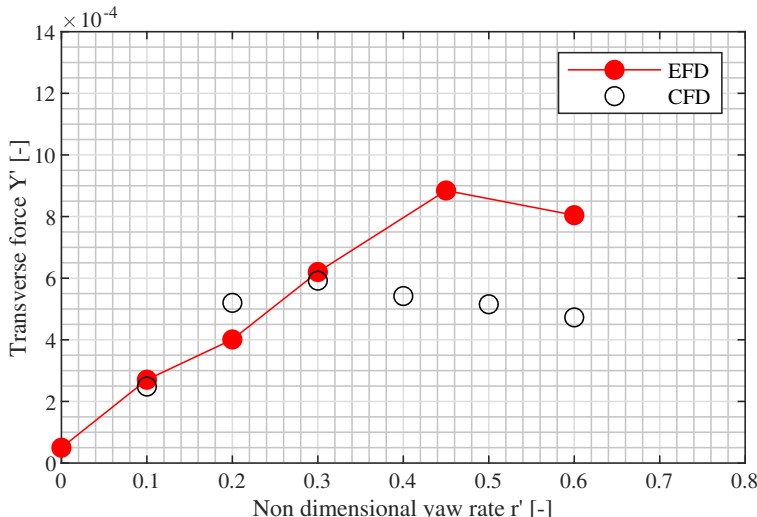

**Figure 23.** Transverse force in the static yaw rate tests-fully appended configuration.

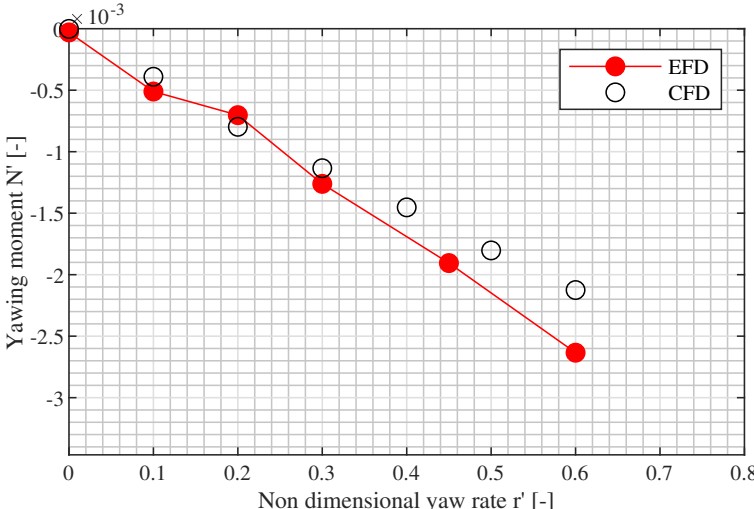

**Figure 24.** Yawing moment in the static yaw rate tests-fully appended configuration.

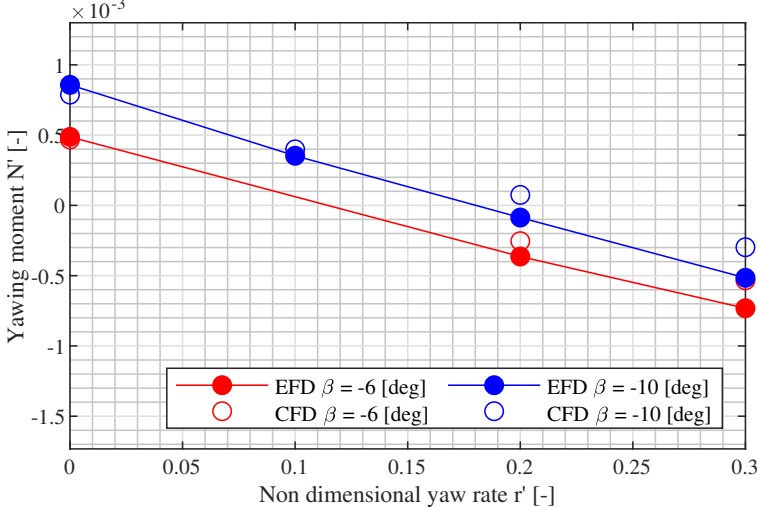

**Figure 25.** Yawing moment in the combined tests-fully appended configuration.

### 4.4. Rudder Forces and Appendages Linearization

Since MARIN was used to measure the rudder forces in the captive tests, this section presents the comparison between the numerical and experimental rudder forces. The rudders are divided as outward and inward consistently from their behaviour during the manoeuvre associated to the specific drift angle or rate of turn. Figure 26 shows that CFD obtained a good prediction of the lateral force of the inward rudder in pure drift tests. On the other hand, the external rudder force was well captured for a drift angle below 12 degrees. The lateral force of the rudder was under-predicted by CFD for higher drift angles.

This phenomenon refers to an anticipation of the stall in the calculations. This is an assessed limit of the application of CFD on the prediction of the rudder performance, and it can refer to the inadequate mesh density on and near the rudders [37]. The stall anticipation could also be ascribed to several aspects related the CFD simulation modelling, such as the actuator disk (propeller outflow) and the turbulence. On the other hand, this study did not deeply investigate the reasons for this phenomenon; however, we relate the stall anticipation with the prediction of the manoeuvrability of the ship in the fully appended configuration.

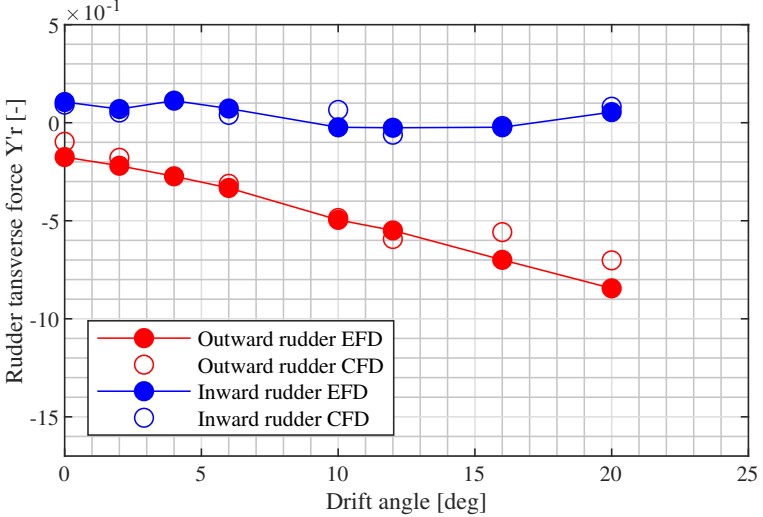

**Figure 26.** Rudder transverse force in the static drift tests.

Since the rudders are the only instrumented appendage in the experimental tests it was not possible to verify if the same behaviour was present also for the other appendages. Figure 27 shows the streamlines near the external rudder for three drift angles (10, 16 and 20 degrees). It is visible that, for the higher angles, a stall phenomenon occurred. This is consistent with the computed force behaviour for the external rudder. The streamlines for the internal rudder are not reported for the sake of brevity, since, in that case, the rudder angles were rather limited—not leading to a stall. This was due to the effect of the hull, which straightened the inflow at the internal rudder location.

Figure 28 shows the lateral forces of the rudders in the pure rotation tests. This figure highlights a higher discrepancy between the CFD and experiments for the external rudder force at higher rates of turn: as in the previous case of the yaw moment, the lateral force of the external rudder was under-estimated for rate of turn beyond 0.3 [-]. In this case, the discrepancy may be ascribed to an anticipation of the stall at the external rudder, as visible in the following Figure 29. Overall the underestimation of the outward rudder force in the drift and rotation tests was around 20%. As shown before for the whole hull forces, an higher discrepancy was present for the forces and moments in the pure yaw tests compared with in the pure drift tests.

This discrepancy may be ascribed to the appendages contribution. The reason that the appendages affected the rotation tests more is given by the following Figure 30. Figure 30 reports the percentage of the hull, appendices and rudders contribution on the global

forces acting on the hull in the pure drift and pure rotation tests. This figure shows that, in the rotation tests, the impact of the rudders and appendices on the global N moment was higher than for the lateral force Y in the pure drift tests. In the pure drift tests, the Y global force was produced 80–85% by the hull and about 15–20% by the appendices and the rudders. In the rotation tests, in contrast, the N global moment was produced 55–65% by the hull and 35–45% by the appendages.

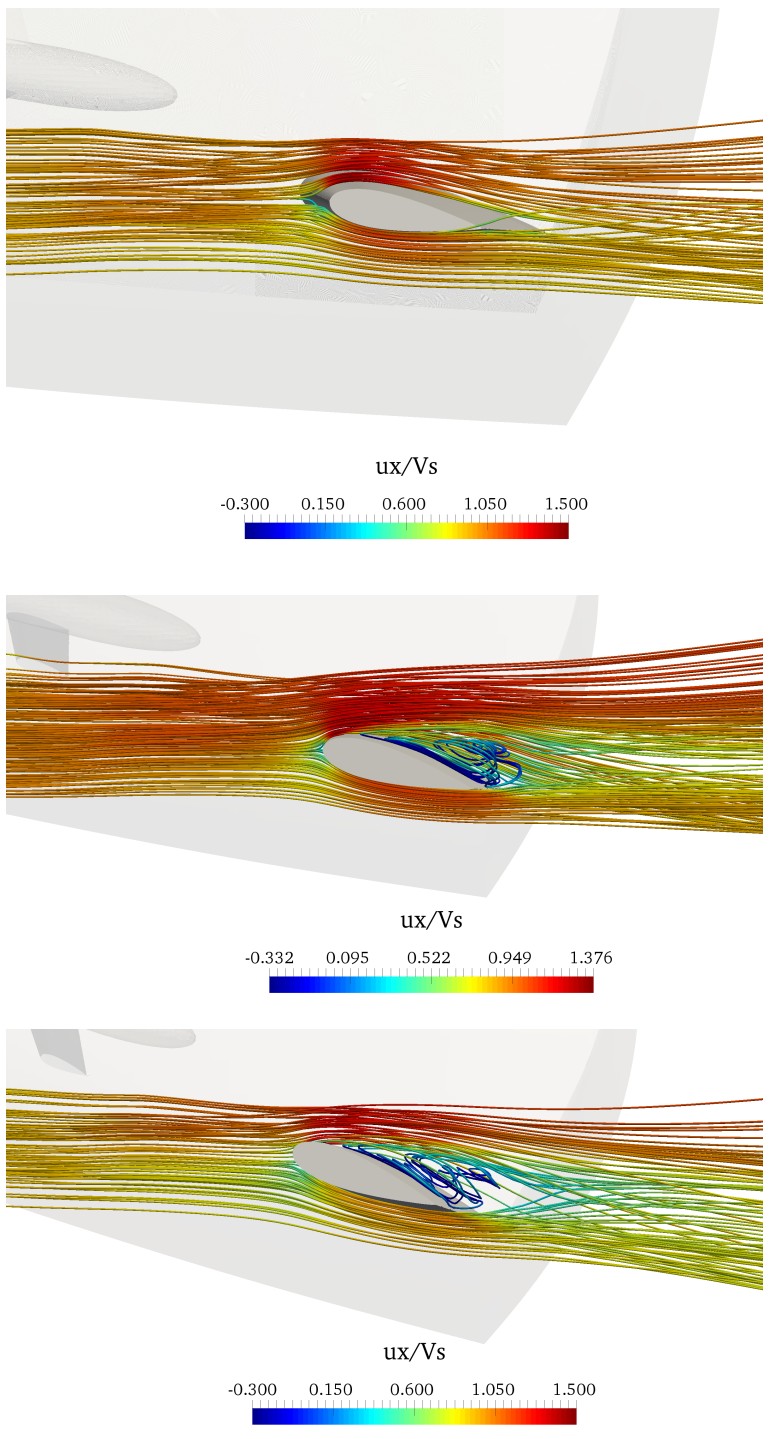

**Figure 27.** Rudder streamlines in the pure drift tests $\beta$ = 10 [deg], $\beta$ = 16 [deg] and $\beta$ = 20 [deg].

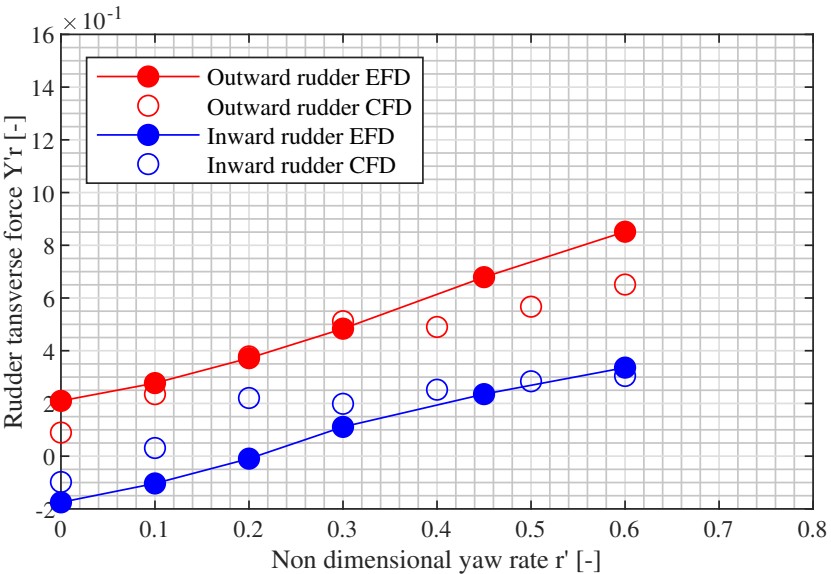

**Figure 28.** Ruddertransverse force in the static yaw rate tests.

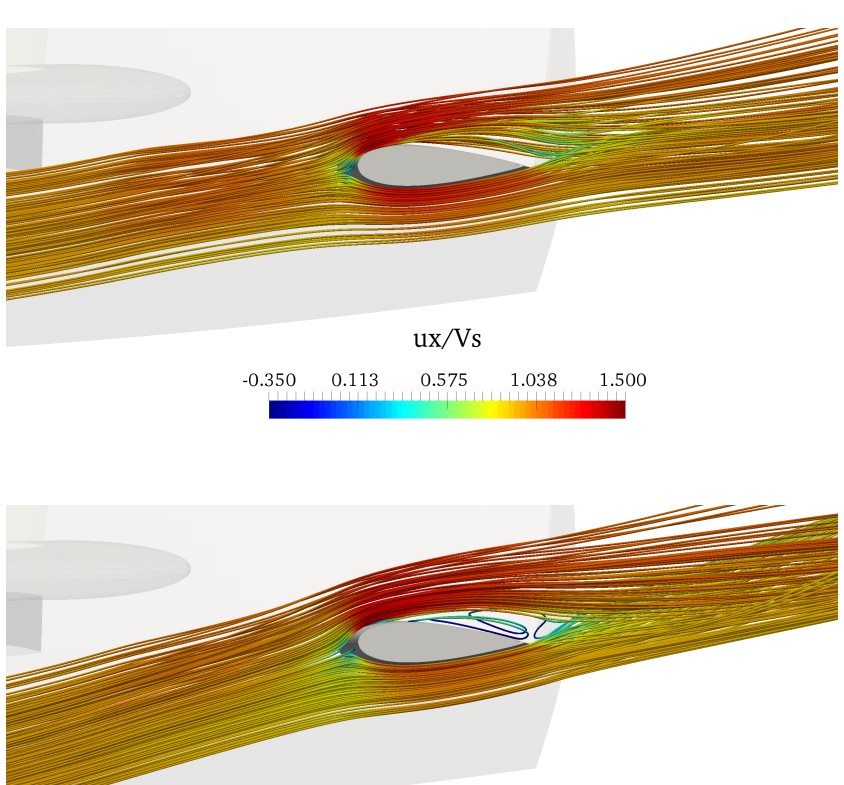

**Figure 29.** Rudder stall in the rotation tests r′ = 0.3 [-] and r′ = 0.6 [-].

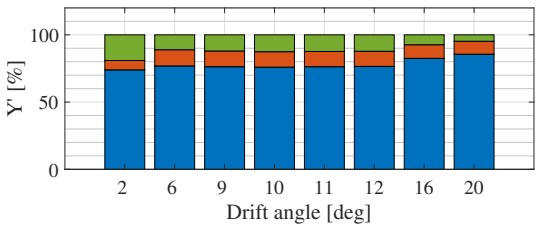

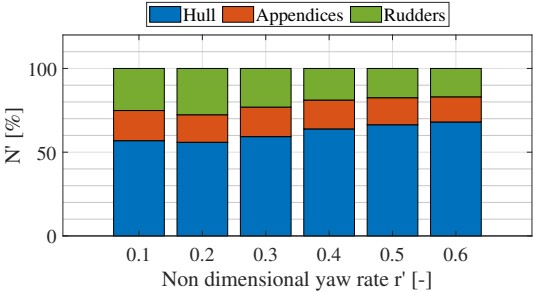

**Figure 30.** Percentage of the global force produced by the hull, appendices and rudders.

Therefore, the under-estimation of the lateral forces on the appendages led to a higher error on the global quantities in the rotation tests compared with in the drift tests.

Since the under-estimation of the outward rudder force was proven by comparison with the experimental measurements, the lateral force of the whole appendages contribution was linearised in order to assess the impact of this under-prediction on the main global quantities. The linearisation of the appendages in rotation was corroborated by the forces obtained through the semi-empirical models of the appendages implemented in the simulator, as presented in Figure 31.

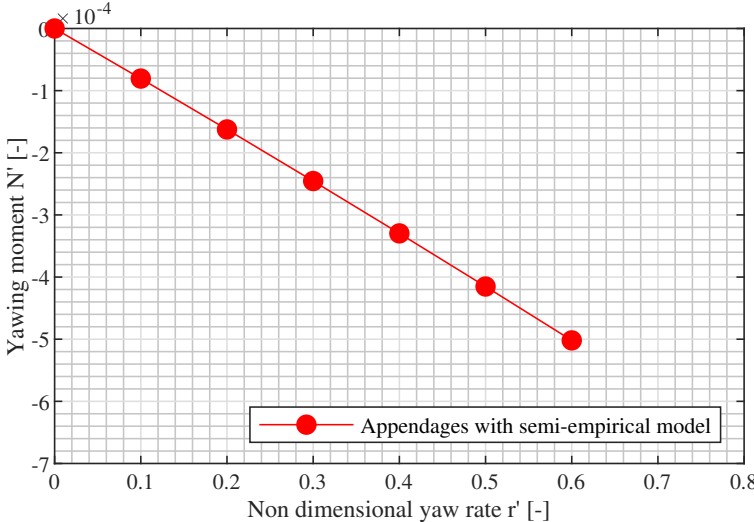

**Figure 31.** N' of the appendages in the rotation tests by means of semi-empirical models.

Figure 32 shows the impact of the linearisation of the appendices contribution in the rotation tests. Triangular marks represent the linearisation of the outward rudder contribution to the yawing moment, which is the lack of description tested by experimental measurements, while the black squared marks represent the global linearisation of the appendages contribution in the pure rotation tests. The linearisation was obtained by analysing the CFD data in the linear range of forces and moments, and then it was extended in the whole range considered accordingly.

In the latter case, the global quantity predicted, with high accuracy, the experimental data, and, even if no measurements are available out of the rudder forces, the under-prediction of the yawing moment could be considered to be related to this phenomenon.

In conclusion, the CFD provided an acceptable prevision of the global forces acting on the hull for the drift and yaw tests; however, it tended to underestimate the forces acting on the various appendages due to an anticipation of stalling phenomena.

This effect resulted in a larger underestimation of the yawing moment in the rotation tests compared with for the lateral force in the drift tests. The local forces acting on the appendages were under-predicted by CFD for higher local angles of attacks (due to $\beta$ or r'). The reason for this under-prediction by means of numerical approaches could be ascribed to a low mesh resolution or other effects connected to the boundary layer description on the lifting surfaces [30].

The analysis of this effect is out of the scope of the present work, which was devoted to the study of approaches affordable in a day-by-day activity during the design process. A complete analysis would require a considerably larger number of cells, leading to very high computational times, which is not acceptable during routine activities. Further analysis will be carried out in the future in order to better understand and quantify the needs, in order to obtain insight into the possible advantages and shortcomings.

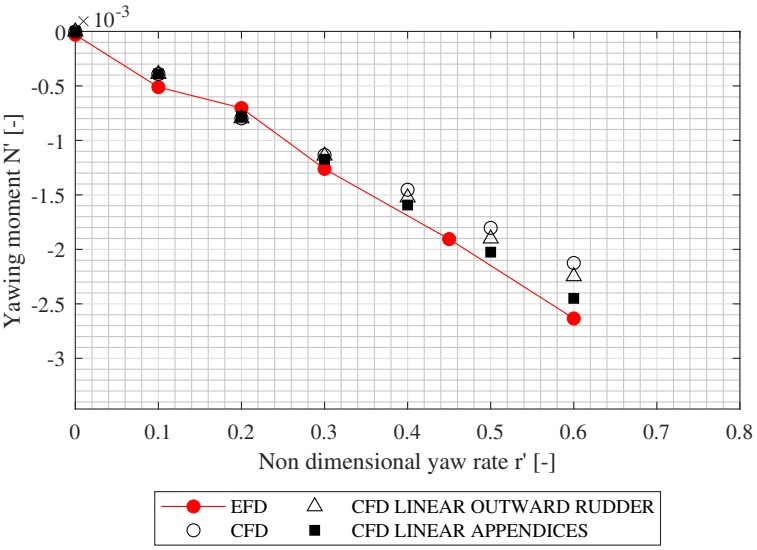

**Figure 32.** N' in the rotation tests with linear contribution of the appendices.

## 5. Manoeuvre Simulations

The results presented in the previous chapter were used to evaluate the hydrodynamic coefficients related to the hull forces. The coefficients computed from the bare hull calculations provide forces and moments acting on the hull alone during the manoeuvres, while the appendices contributions were included using regression formulae developed in previous activities. On the contrary, in the case of calculations for the appended hull, global forces were introduced directly, including the appendages contribution. For these simulations, only the rudder effect was modelled by the previous mentioned mathematical approaches.

Table 13 reports the hydrodynamic coefficients computed from both the bare hull and the fully appended. The linear coefficients reported in this table are consistent with the appendages contribution during the manoeuvres: $Y_v$ related to the fully appended case was higher while the $N_v$ coefficient was lower (a higher lateral force at stern and more stable ship), $N_r$ was higher in the fully appended case with respect to bare hull, which leads to a more stable ship. Table 13 also includes the added masses considered in the simulations. The added masses were the same for bare hull and fully appended calculations, and they were computed according to statistical formulations. The manoeuvres were simulated with an approach speed equal to 18 knots as in the free running model tests.

**Table 13.** The numerical hydrodynamic coefficients at 18 knots.

| Hydrodynamic Coefficients | | |
|---|---|---|
| Configuration | Bare Hull | Appended |
| Added masses | | |
| $X_{\dot{u}}$ [-] | $-3.53 \times 10^{-4}$ | |
| $Y_{\dot{v}}$ [-] | $-4.41 \times 10^{-3}$ | |
| $Y_{\dot{r}}$ [-] | $0$ | |
| $N_{\dot{v}}$ [-] | $0$ | |
| $N_{\dot{r}}$ [-] | $-2.76 \times 10^{-4}$ | |
| Longitudinal force X' | | |
| $X_{vv}$ [-] | $-5.17 \times 10^{-3}$ | $-5.71 \times 10^{-3}$ |
| $X_{rr}$ [-] | $-1.17 \times 10^{-3}$ | $-1.31 \times 10^{-3}$ |
| $X_{vr}$ [-] | $6.21 \times 10^{-4}$ | $8.57 \times 10^{-4}$ |
| Transverse force Y' | | |
| $Y_v$ [-] | $-1.17 \times 10^{-2}$ | $-1.10 \times 10^{-2}$ |
| $Y_r$ [-] | $-3.21 \times 10^{-3}$ | $3.50 \times 10^{-4}$ |
| $Y_{v\|v\|}$ [-] | $0$ | $-3.17 \times 10^{-2}$ |
| $Y_{vvv}$ [-] | $-7.30 \times 10^{-2}$ | $0$ |
| $Y_{r\|r\|}$ [-] | $0$ | $-1.68 \times 10^{-3}$ |
| $Y_{rrr}$ [-] | $-6.39 \times 10^{-4}$ | $0$ |
| $Y_{vvr}$ [-] | $-3.64 \times 10^{-2}$ | $1.95 \times 10^{-3}$ |
| $Y_{vrr}$ [-] | $-2.05 \times 10^{-2}$ | $-1.49 \times 10^{-3}$ |
| Yawing moment N' | | |
| $N_v$ [-] | $-6.50 \times 10^{-3}$ | $-6.00 \times 10^{-3}$ |
| $N_r$ [-] | $-1.79 \times 10^{-3}$ | $-2.79 \times 10^{-3}$ |
| $N_{v\|v\|}$ [-] | $-5.20 \times 10^{-3}$ | $-2.31 \times 10^{-3}$ |
| $N_{vvv}$ [-] | $0$ | $0$ |
| $N_{r\|r\|}$ [-] | $0$ | $-7.84 \times 10^{-5}$ |
| $N_{rrr}$ [-] | $-1.73 \times 10^{-3}$ | $0$ |
| $N_{vvr}$ [-] | $-2.13 \times 10^{-2}$ | $-2.86 \times 10^{-2}$ |
| $N_{vrr}$ [-] | $-5.15 \times 10^{-3}$ | $-7.52 \times 10^{-3}$ |

The tables reported below show the main characteristics of the zig-zag $10°/10°$, zig-zag $20°/20°$ and turning circle $35°$ manoeuvres performed with the adoption of the hydrodynamic coefficients reported in Table 13. For the sake of completeness, the available time traces of the manoeuvres are reported in the figures below: these time traces are referred to a single free running test, whereas the tabular values were computed by the average for the model tests.

Table 14 shows that the simulations led to a substantial agreement with the free running model test results with discrepancies in the overshoot angles lower than 0.3 degrees. The hydrodynamic coefficients computed from the bare hull and appended configuration produced similar results in terms of the angles and times of the manoeuvre. The time traces of the free running model tests related to the zig-zag $10°/10°$ are not available, and only the tabular comparison is reported.

Figure 33 shows the numerical and the experimental trajectories related to the zig-zag $20°/20°$ manoeuvre quantitatively reported in Table 15. The high quality of the prevision of the first overshoot angle is shown, and the second and the third overshoot angles are described with an error lower than 1 degree. For the zig-zag $20°/20°$ manoeuvre, the appended configuration led to results slightly closer to the experiments than the bare hull

configuration. Figure 34 and Table 16 report the main characteristics of the turning circle 35° manoeuvre graphically and as a table, respectively.

**Table 14.** The numerical simulation and experimental results of the zig-zag $10°/10°$ manoeuvre.

| Zig-Zag $\delta = 10°$ | | | | | |
|---|---|---|---|---|---|
| Configuration | EFD | CFD bare hull | | CFD appended | |
| | Value | Value | Error [%] | Value | Error [%] |
| 1st $OV_a$ [deg] | 2.1 | 1.84 | −14.1 | 1.85 | −13.3 |
| 1st $OV_t$ [s] | 7.50 | 6.00 | −25.0 | 6.10 | −23.0 |
| 2nd $OV_a$ [deg] | 2.50 | 2.17 | −15.3 | 2.19 | −14.4 |
| 2nd $OV_t$ [s] | 6.90 | 6.80 | −1.5 | 6.90 | −0.1 |
| 3rd $OV_a$ [deg] | 2.50 | 2.17 | −15.1 | 2.19 | −14.3 |
| 3rd $OV_t$ [s] | 7.50 | 6.70 | −11.9 | 6.90 | −8.7 |
| OV period [s] | 110.1 | 103.8 | −6.1 | 104.8 | −5.1 |
| 2nd execution time [-] | 1.29 | 1.59 | 18.9 | 1.60 | 19.2 |

**Table 15.** The numerical simulation and experimental results of the zig-zag $20°/20°$ manoeuvre.

| Zig-Zag $\delta = 20°$ | | | | | |
|---|---|---|---|---|---|
| Configuration | EFD | CFD bare hull | | CFD appended | |
| | Value | Value | Error [%] | Value | Error [%] |
| 1st $OV_a$ [deg] | 4.80 | 4.53 | −6.0 | 4.60 | −3.7 |
| 1st $OV_t$ [s] | 7.90 | 7.30 | −8.2 | 7.30 | −8.2 |
| 2nd $OV_a$ [deg] | 5.70 | 4.78 | −19.3 | 4.90 | −16.1 |
| 2nd $OV_t$ [s] | 8.60 | 7.60 | −13.2 | 7.60 | −13.2 |
| 3rd $OV_a$ [deg] | 3.80 | 4.77 | 20.3 | 4.90 | 22.4 |
| 3rd $OV_t$ [s] | 8.60 | 7.50 | −14.7 | 7.70 | −11.7 |
| OV period [s] | 114.6 | 113.9 | −0.6 | 113.3 | −1.1 |
| 2nd execution time [-] | 1.66 | 1.66 | 0.4 | 1.66 | −0.1 |

As expected, fully appended calculations led to a more unstable ship in the turning circle simulation (e.g., a smaller tactical diameter). This lack of description can be related to the under-estimation of the global yawing moment at larger values of r′ in the appended configuration as reported in Section 4. The discrepancies between the simulations and the experiments in the manoeuvre trajectories cannot be related only to the hull and the appendages modules; they are influenced by the rudder and the propeller action.

MARIN was used to measure the rudder forces and propeller thrusts during the free running model tests. Therefore, the comparison between the experimental measurements and simulations time traces are reported in Figures 35 and 36 in terms of the total propeller thrust and total rudder lateral force. These figures show that the propellers thrust was under-predicted by the simulator mathematical model during the zig-zag $20°/20°$, while it showed good agreement with the experiments in the turning circle. On the other hand, the rudders lateral force was accurately predicted in the zig-zag $20°/20°$, while it was over-predicted during the turning.

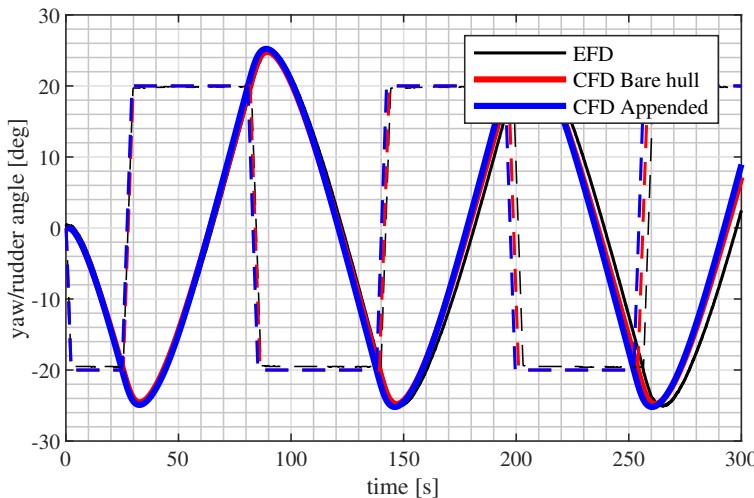

**Figure 33.** Zig-zag $\delta = 20°$ trajectory.

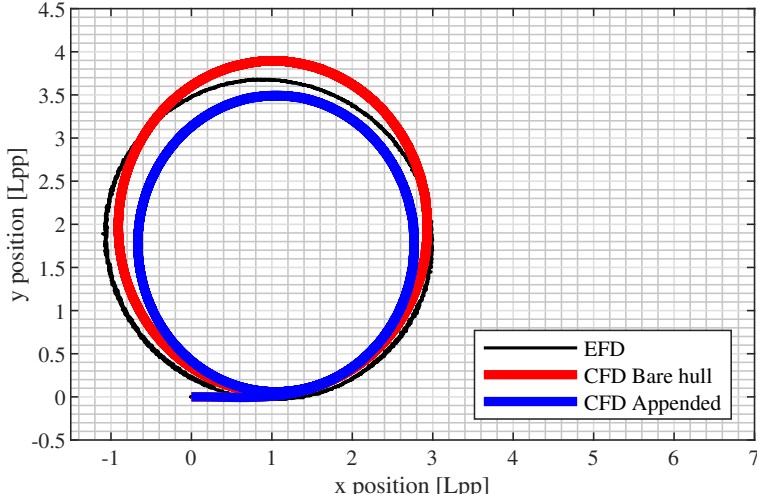

**Figure 34.** Turning circle $\delta = 35°$ trajectory.

**Table 16.** The numerical simulation and experimental results of the turning circle $\delta = 35°$ manoeuvre.

| | | | | | |
|---|---|---|---|---|---|
| **Turning Circle $\delta = 35°$** | | | | | |
| Configuration | EFD | CFD bare hull | | CFD appended | |
| | Value | Value | Error [%] | Value | Error [%] |
| AD [$L_{PP}$] | 2.95 | 2.88 | −2.4 | 2.71 | −8.7 |
| TR [$L_{PP}$] | 1.61 | 1.54 | −4.2 | 1.39 | −15.6 |
| TD [$L_{PP}$] | 3.96 | 3.84 | −3.1 | 3.47 | −14.0 |
| FD [$L_{PP}$] | 3.80 | 3.84 | 1.0 | 3.45 | −10.0 |
| r [deg/s] | 1.47 | 1.47 | 0.2 | 1.60 | 7.5 |
| $\beta$ [deg] | −11.7 | −12.9 | 9.5 | −13.5 | 13.4 |
| T90 [s] | 67.0 | 65.6 | −2.1 | 61.6 | −8.8 |
| T180 [s] | 120.0 | 126.6 | −1.9 | 118.1 | −9.2 |
| T360 [-] | 255.0 | 248.8 | −2.5 | 231.3 | −10.2 |

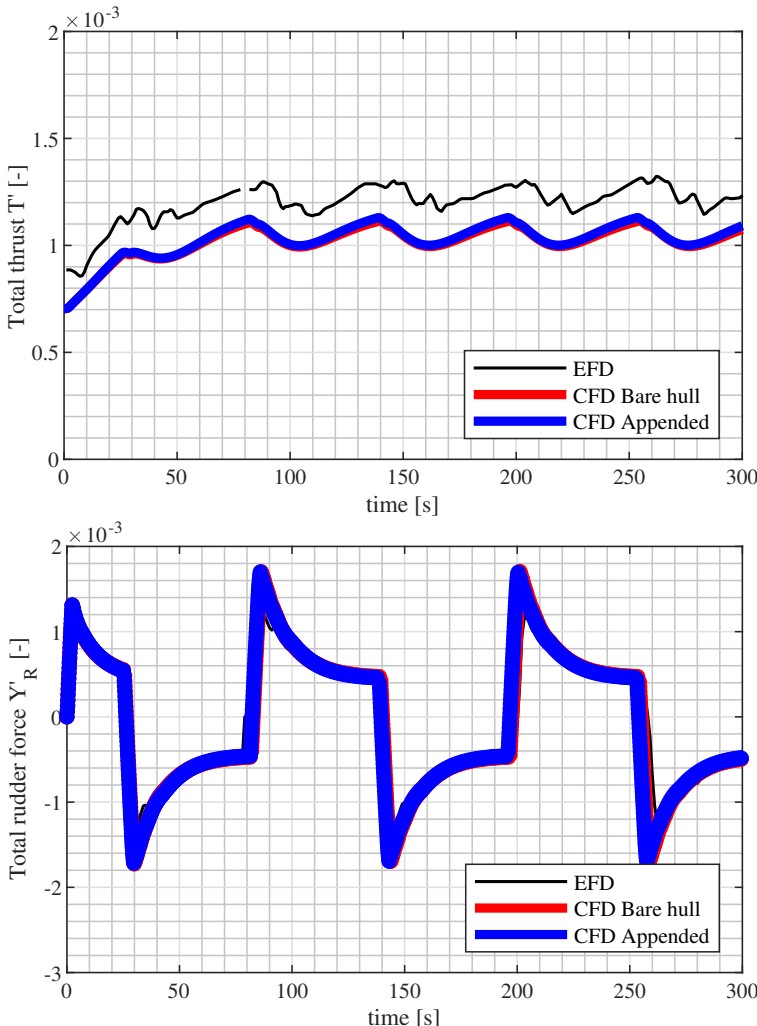

**Figure 35.** Thrust and rudder total forces time traces in zig zag 20/20.

The conclusion is that the bare hull configuration led to results closer to the free running measurements in the turning circle manoeuvres; however, this prediction is slightly affected by the discrepancies in the rudder and the propeller behaviour during the manoeuvres. Looking to Figure 36, since the rudders contribution was over-predicted in the simulator, the fully appended configuration led to a more unstable ship due to this additional destabilizing force.

On the other hand, the rudder contribution in the zig-zag 20°/20° manoeuvre showed very good agreement to the experiments even if the propeller thrust showed a slight difference with the experiments. Nevertheless, this under-prediction had a slight influence on the total lateral force of the rudders (about 5%), and thus a negligible influence on the manoeuvre is expected.

This study did not investigate the rudder and the propeller mathematical models adopted in the simulator; however, these considerations lead to the conclusion that the simulation results could be affected by these modules, which will be further analysed. The differences between the experiments and the simulations could be related to the rudder and propeller mathematical models, the straightening coefficients, the wake fraction at the propellers (e.g., the function of the drift angle instead of constant values) and the propeller–rudder interaction factor. All these aspects will be studied in future activities. The simulator provides the possibility of computing the hydrodynamic coefficients of the hull module by means of semi-empirical mathematical models.

These are (i) a model suitably developed by DITEN over years starting from Ankudinov regressions [38], modified and customized for twin screw ships, with the capability

to capture the appendages effect [10]; and (ii) a model based on the strip theory approach [39–41], suitably tuned on experimental results of twin screw ships. The particulars of these models are not discussed within this study; however, the simulation results obtained are reported and compared with the CFD manoeuvres to demonstrate the improvement of the manoeuvrability prediction with the computation of the hydrodynamic coefficients with the CFD code.

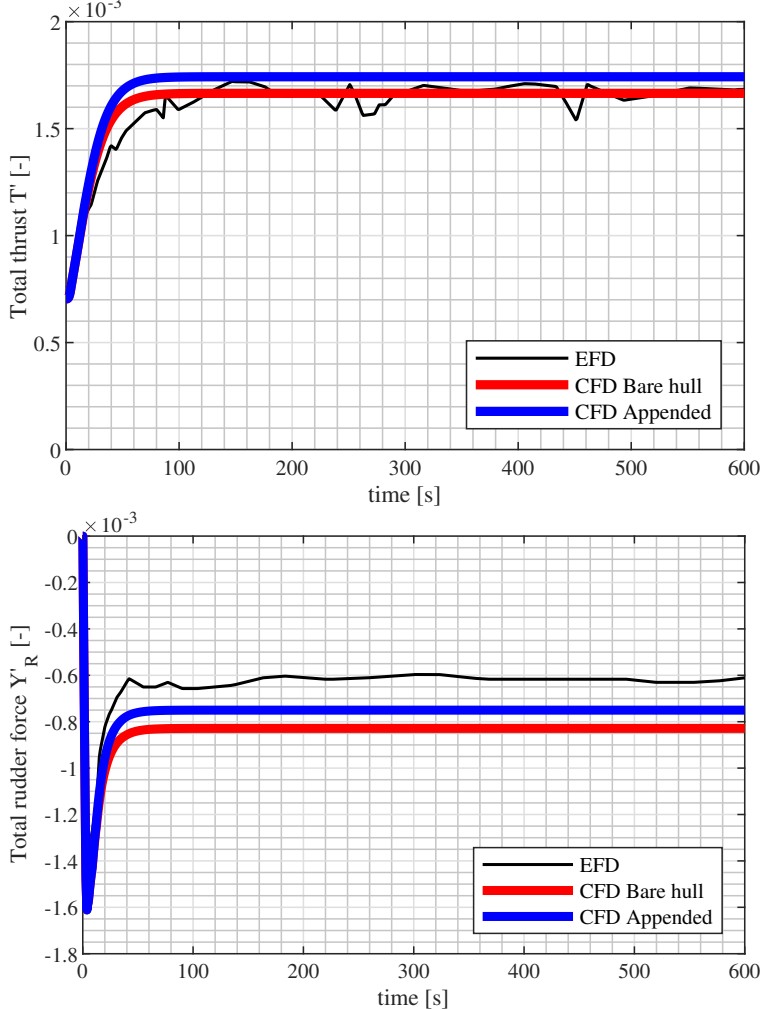

**Figure 36.** The thrust and rudder total force time traces in a turning circle.

In addition to the semi-empirical calculations, a further simulation was conducted using the hydrodynamic coefficients computed from the yawing moment corrected with the linearised contribution of the appendices in the pure rotation tests, as reported in Section 4.

Figure 37 reports graphically the main manoeuvring characteristics of these sets related to the zig-zag 10°/10° manoeuvre. This figure shows that the CFD simulations predicted, with higher accuracy, the overshoot angles with respect to the semi-empirical models. Moreover, considering the appendage correction (violet bar), the results are closer to the experimental ones. Figure 38 shows the comparison between the overshoot angles related to the zig-zag 20°/20° manoeuvres.

The high variability of the overshoot angles in free running model tests did not permit us to identify the best setup in order to predict the characteristics of the manoeuvre. Figure 39 reports the mean values of the three overshoot angles. This figure shows that all the numerical approaches provided an accurate estimate of the average values of the overshoot angles, while the semi-empirical methods led to a slight under-prediction of the angle (difference lower 0.5 degrees).

Regarding the turning circle manoeuvre, Figure 40 shows that the setup that led to the closest prediction of the free running tests was the numerical setup for the bare hull calculations. Semi empirical approaches led to a good agreement of the manoeuvre characteristics, whereas the appended numerical setup was affected by the instability due to the under-estimation of the yawing moment in rotation tests. The linearisation of the appendices contribution improved the prediction of the manoeuvre, partially recovering the differences between the bare and the appended configuration.

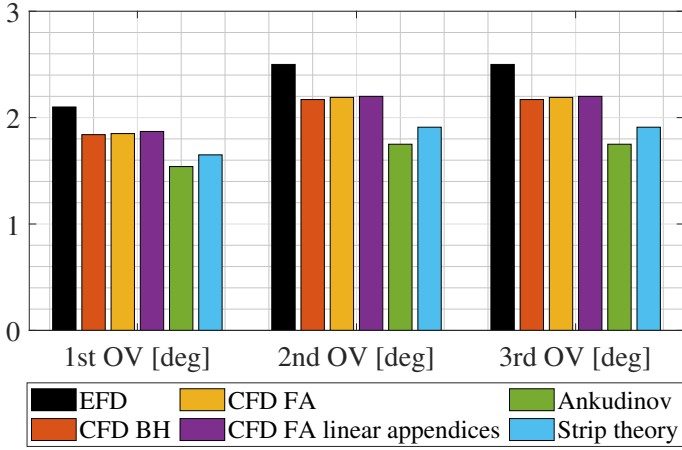

**Figure 37.** The Zig-Zag 10°/10° manoeuvre characteristics.

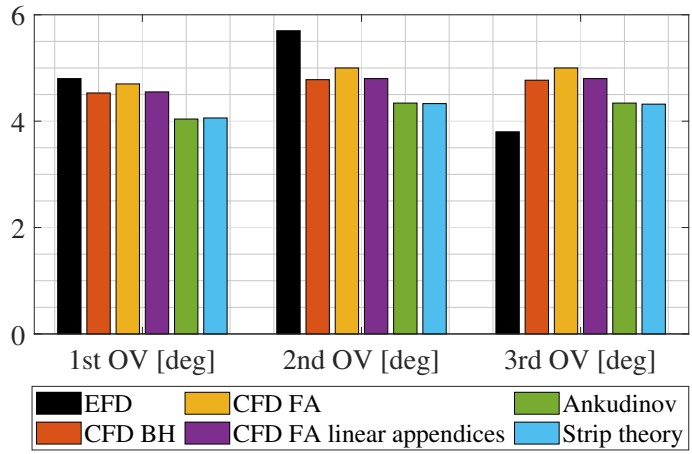

**Figure 38.** The Zig-Zag 20°/20° manoeuvre characteristics.

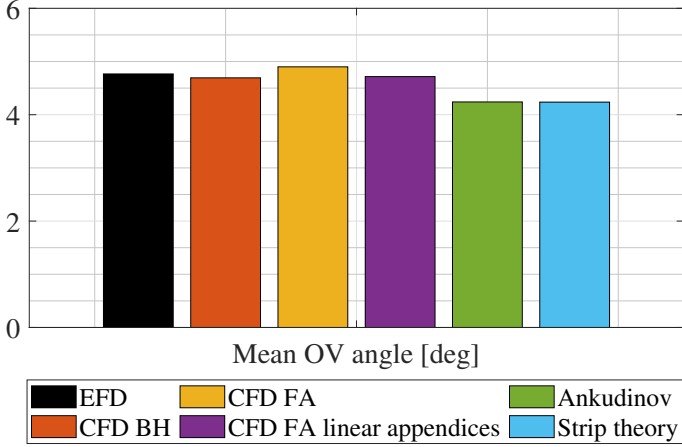

**Figure 39.** The Zig-Zag 20°/20° mean overshoot angles.

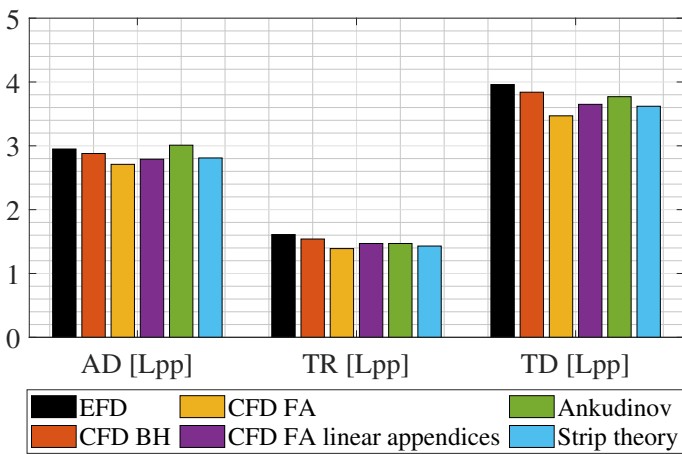

**Figure 40.** The Turning circle $\delta = 35°$ manoeuvre characteristics.

## 6. Conclusions

In this study, we investigated the whole manoeuvre simulation chain from captive test simulations with numerical approaches to time domain simulations of the manoeuvres. The CFD calculations were conducted on both the bare hull and the appended configurations. The numerical results were compared with the available experimental measurements. The bare hull CFD calculations demonstrated the high quality of the predicted global forces acting on the hull. On the other hand, the appended setup led to good agreement with the experiments in the pure drift tests and to an under-prediction of the yawing moment in the rotation tests.

Considering the results in terms of the rudder forces, it is possible to ascribe a large part of the above mentioned discrepancy to an anticipated stall of the appendages in correspondence with the high drift and yaw rate values. The linearisation of the appendages contribution led to an increase of the accuracy of the yawing moment with respect to the experiments.

The manoeuvre simulations were in line with the computed force results: the bare hull hydrodynamic coefficients led to manoeuvre characteristics close to the free running model tests (this result was also used to assess the quality of the appendices mathematical models that were implemented in the simulator), while the appended hull hydrodynamic coefficients resulted in a more unstable ship even if the results could be considered globally satisfactory. The linearisation of the appendices contribution led to an improvement of the prevision of the manoeuvre characteristics.

In brief, this paper provides a guideline regarding a numerical procedure to assess the manoeuvring ability of a ship in the first design stages. The use of a quasi-steady simulator in combination with a proper running attitude modification model led to good results in terms of the global forces acting on the ship in the captive model tests. The CFD calculations performed on the appended configuration of the ship indicated certain difficulties regarding the description of the appendices contribution—in particular, in the rotation tests (as these tests are highly influenced by the appendices behaviour).

These difficulties must be discussed and further investigated; however, they are not the subject of this paper. The simulations showed satisfactory results in terms of prediction of the main characteristics of the manoeuvres of the ship. Numerical hydrodynamic coefficients in combination with semi empirical models for the description of the rudder–propeller contributions led to better results with respect to the semi-empirical approaches.

**Author Contributions:** Conceptualization, M.V., D.V., and R.T.; methodology, M.V., D.V., B.P. and A.F.; software, D.V., B.P. and A.F.; M.V., D.V. and A.F.; formal analysis, M.V., D.V., B.P. and R.T.; investigation, D.V. and A.F.; resources, M.V., D.V. and A.F.; data curation, A.F.; writing—original draft preparation, M.V., D.V. and A.F.; writing—review and editing, M.V., D.V., B.P., R.T. and A.F.;

visualization, A.F.; supervision, M.V., D.V. and R.T.; project administration, M.V. and D.V. All authors have read and agreed to the published version of the manuscript.

**Funding:** This research is partially funded by the Regione Liguria administration.

**Institutional Review Board Statement:** Not applicable.

**Informed Consent Statement:** Not applicable.

**Data Availability Statement:** The experimental results shown in this paper were received from SIMMAN conferences held in 2008 and 2014. Data are made available from conference registration and attendance. The sites of the events are shown below. The data reported are not in any open-databases.[http://www.simman2008.dk/] [https://simman2014.dk/].

**Acknowledgments:** The Authors wish to express their gratitude to Mr. Jacopo Labanti (Fincantieri MM-ARC office) for his constant support throughout this project.

**Conflicts of Interest:** The authors declare no conflict of interest.

## Nomenclature

| | |
|---|---|
| CFD | Computational Fluid Dynamics |
| EFD | Experimental Fluid Dynamics |
| MMG | Manoeuvring Modeling Group |
| SOLAS | Safety of Life at Sea |
| IMO | International Maritime Organization |
| RANS | Reynolds Average Navier-Stokes |
| DOF | Degrees of Freedom |
| $\lambda$ | Scale factor |
| $L_{PP}$ | Length between perpendiculars |
| $A_{WL}$ | Waterline area |
| $B$ | Beam |
| $T$ | Draught |
| $\nabla$ | Ship volume |
| $\Delta$ | Ship displacement |
| $I_{ZZ}$ | Second moment of area of the waterplane |
| $x_G$ | Longitudinal position centre of gravity |
| $x_B$ | Longitudinal position centre of buoyancy |
| $GM_T$ | Transverse metacentric heigth |
| $R_L$ | Longitudinal metacentric radius |
| $\beta$ | Drift angle |
| $\delta$ | Rudder angle |
| $\phi$ | Roll angle |
| $\theta$ | Pitch angle |
| $\Psi$ | Yaw angle |
| R | Ship resistance |
| $X$ | Longitudinal force |
| K | Roll moment |
| $Y$ | Transversal force |
| $M$ | Pitch moment |
| $Z$ | Vertical force |
| $N$ | Yawing moment |
| u | Surge velocity |
| p | Roll velocity |
| v | Sway velocity |
| q | Pitch velocity |
| w | Heave velocity |

| | |
|---|---|
| r | Yaw rate |
| V | Ship velocity |
| Fr | Froude number |
| $x_R$ | Longitudinal position of rudder |
| $\gamma_{vR}, \gamma_{rR}$ | Straightening coefficients at rudder |
| $s_R$ | Rudder span |
| $t_R$ | Rudder thickness |
| $c_R$ | Rudder chord |
| $C_L, C_D$ | Rudder lift and drag coefficient |
| $x_P$ | Longitudinal position of propeller |
| $\gamma_{vP}, \gamma_{rP}$ | Straightening coefficients at propeller |
| $D_P$ | Propeller diameter |
| $C_{0.75}$ | Blade chord at 0.7r/R |
| $A_E/A_0$ | Blade area ratio |
| $Z_P$ | Number of propeller blades |
| $rpm$ | Revolutions per minute |
| $\nu$ | Kinematic viscosity |
| $w_P$ | Wake fraction at propeller |
| $t_P$ | Thrust deduction factor |
| $\rho$ | Fluid density |
| $g$ | Gravity acceleration |
| $AD$ | Advance |
| $TR$ | Transfer |
| $TD$ | Tactical diameter |
| $FD$ | Final diameter |
| $OV_a$ | Overshoot angle |
| $OV_t$ | Overshoot time |
| $LTS$ | Local Time Stepping |
| $VOF$ | Volume of Fluid |
| $SST$ | Menter's Shear Stress Transport |
| UNIGE | Universitá Degli Studi Di Genova |
| BH | Bare Hull |
| FA | Fully Appended |

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
