# Peer review of "Assessment of the Manoeuvrability Characteristics of a Twin Shaft Naval Vessel Using an Open-Source CFD Code"

_jmse, doi:10.3390/jmse9060665_

Round 1

Reviewer 1 Report

Effective modelling of ship motion is an important issue that continues to grow in importance with increasing automation in shipping. Finding a compromise between effort and benefit, for example in terms of mesh density in CFD, is an important contribution in this modelling. The simulations carried out and the comparison with actual experimental measurements form a good basis for an evaluation of the open-source application. Unfortunately, the paper lacks systematics and scientific comparison to classical modelling methods in efficiency and accuracy. 

Generally necessary changes:

  • Frequently colloquial phrases
  • Too long sentences with many repetitions or less relevant information
  • Single sentences between illustrations should be avoided, otherwise it is difficult for the reader to establish the context in the text. 
  • Line numbering is interrupted 
  • consistent wording throughout the paper, e.g. manoeuvrability instead of manoeuvring ability
  • Given the length of the paper, the introduction should provide a concise overview of which topics are covered and in what systematic order. 

Author Response

Dear Sir/Madam

Please see the attachment. You will find the point-by-point response to your valuable review.

Yours faithfully,

Andrea Franceschi

Reviewer 2 Report

Thanks to the Authors for this interesting paper that talks about the assessment of the quality of maneuvers prediction of the DTMB 5415 by means of a time-domain simulator based on CFD hydrodynamic coefficients. The hydrodynamic coefficients of the hull in the bare or appended configuration are computed by virtual captive tests carried out using the CFD open-source code OpenFOAM.

The paper talks about a well-known and well-investigated topic. However, the paper is all in all interesting. There are a few points that need to be clarified.

Following my comments.

As a general comment, I would recommend the Authors use the new abbreviation for “Froude Number” (Fr) instead of Fn (the old one).

Numerical setup

I would suggest adding the details of the domain dimensions (shown in table 1) in Figure 4. Indeed, would be easier to understand the whole dimensions of the domain.

Figure 6. As far as I understood the domain for the rotation test simulation is different from the pure drift test. In this case, I would suggest describing geometrically both of them.

Which kind of mesh approach for the hull motion has been used? Is not clearly exposed in the text.

Table 2. “Richardson mesh convergence analysis”. This table shows the results of a sort of mesh independence Analysis. The Authors quote in the text the study of Celik et al., 2008. However, the “Richardson mesh convergence analysis” exposed is not in the line with what indicated by the Celik et al. approach. I would suggest reshaping this part of the paragraph exposing the results in a way similar to the typical Verification analysis (take a look at the ITTC Uncertainty Analysis in CFD Verification and Validation Methodology and Procedures 7.5-03 -01-01). Specifically, I would recommend, for instance, for each case, expose the ratio of convergence and indicate if the convergence is monotonic, oscillatory, etc. (asymptotic, instead, is a non-clear expression in the light of a Verification analysis).

Line 112. “Mesh dependence analysis”. Please check it, generally, this analysis is called “Mesh Independence Analysis”

Figure 10. Please add a legend of the colors in the plot.

Author Response

(The authors gave the same response as above.)

Reviewer 3 Report

  1. The introduction is too short and not well presented. A more comprehensive literature review is expected. Also, the contribution of current work in comparison to the existing researches should be highlighted explicitly.
  2. The vertical axis defined in Figure 1 does not match the description in Section 2.
  3. The font sizes in all figures and tables should be adjusted. It looks like they are not in the right sizes now.
  4. Many symbols used in Equation 2 are not defined in the Nomenclature. Also, I am not sure how equation 2 takes into account 'the influence of roll motion on the in-plane forces and moments,' as claimed by the authors.
  5. I am unable to understand this line below Figure 2 - 'so all the other hydrodynamic devices are modelled using literature semi-empirical mathematical models.' Please clarify.
  6. Figure 5 and 6 appears earlier than Figure 4 in the text. Table 1, 5, Figure 9 etc are not mentioned in the text. The manuscript should be thoroughly checked to rectify all such mistakes. 
  7. In fact, I suggest the entire section 3 should be revised and rewritten. I believe it can be better organized to improve the readability. 
  8. Below table 8, it was mentioned that 'Rotation and combined tests reported in table 11 are performed dynamically, the related forces are computed by means of the procedures provided by the institutes.' Please explain what is the 'means of the procedures provided by the institutes.'
  9.  Section 4 and 5 should be revised to improve the readability of the manuscripts. Besides, some parts of these sections contain several small paragraphs which could be linked. 

Author Response

(The authors gave the same response as above.)

Round 2

Reviewer 2 Report

Dear Authors,

thanks for the revision, the manuscript has been improved after this stage of revision. Personally, I don't have further comments. The paper can be accepted.

Reviewer 3 Report

Thank you for revising the manuscript based on the suggestions from the reviewers. I am satisfied with the new version and have no further comments.